# Development and application of a street-level meteorology and pollutant tracking system (S-TRACK)

Huan Zhang[1,2], Sunling Gong[1]*, Lei Zhang[1]*, Jianjun He[1], Yaqiang Wang[1], Lixin Shi[2,3], Jingyue Mo[1], Huabing Ke[1], Shuhua Lu[1]

[1] State Key Laboratory of Severe Weather & Key Laboratory of Atmospheric Chemistry of CMA, Chinese Academy of Meteorological Sciences, Beijing 100081, China
[2] Key Laboratory of Meteorology and Ecological Environment of Hebei Province, Shijiazhuang 050000, China
[3] Meteorological Institute of Hebei Province, Shijiazhuang 050000, China

*Correspondence to*: Sunling Gong (gongsl@cma.gov.cn) and Lei Zhang (leiz09@cma.gov.cn)

**Abstract.** A multi-model simulation system for street level circulation and pollutant tracking (S-TRACK) has been developed by integrating the Weather Research and Forecasting (WRF), the STAR-CCM+ (Computational Fluid Dynamics model - CFD) and the Flexible Particle (FLEXPART) models. The winter wind environmental characteristics and the potential impact of traffic sources on nearby receptor sites in a city district of China are analysed with the system for January 2019. It is found that complex building layouts change the structure of the wind field and thus have an impact on the transport of pollutants. The wind speed inside the building block is smaller than the background wind speed due to the dragging effect of dense buildings. Ventilation is better when the dominant airflow is in the same direction as the building layout. Influenced by the building layout, the local circulations show that the windward side of the building is mostly the divergence zone and the leeward side is mostly the convergence zone, which is more obvious for high buildings. With the hypothesis that the traffic sources are uniformly distributed on each road and with identical traffic volumes, the potential contribution ratios of four traffic sources to certain specific sites under the influence of the street-level circulations are estimated with the method of residence time analysis. It is found that the contribution ratio varies with the height of the receptor site. As a result of the generally upward motion in the airflow, the

position with the greatest potential contribution ratio from the four road traffic sources is located on a certain height (about 15m in this study). The potential contribution of a road to one of the receptor sites is also investigated under different wind directions. The established system and the results can be used to understand the characteristics of urban wind environment and to help the air pollution control planning in urban areas.

## 1. Introduction

In recent decades, with the continuous development of urban construction in China, urban environmental problems have become increasingly serious and attracted widespread attentions. According to the 2019 China Ecological Environment Status Bulletin, 180 of 337 cities at the prefecture level exceeded ambient air quality standards. The complex building layouts and differences in thermal structures within cities lead to extremely complicated meteorological characteristics and pollutant transport in urban areas (Lei et al., 2012; Fernando et al., 2010; Aynsley, 1989). Though the transport of atmospheric pollution in urban areas is widely studied, the study on tracking the sources of pollutants on the street-level is still lacking due to limitations in research methods.

Researches on the street-level atmospheric environment are mainly divided into three methods: field measurements (Macdonald et al., 1997), laboratory simulation research (Mavroidis et al., 2003), and model simulations (Steenburgh et al., 2015; Hendricks et al., 2007; Yucong et al., 2014). The model simulation has become one of the main methods for studying environmental problems at the street-level due to the easy control of simulation conditions and simple processing steps. The Computational Fluid Dynamics (CFD) is a numerical simulation method to study the fluid thermal-dynamic problems and

is now widely used in the studies related to microscale problems within the urban canopy (Gosman, 1999). The core of CFD simulation method is to solve the Navier-Stokes equations. Depending on the turbulence closure scheme, CFD pre-processing models can be divided into three types: Direct numerical simulation (DNS), Reynolds-averaged Navier–Stokes (RANS) (Liu et al., 2018; Zheng et al., 2015; Milliez and Carissimo, 2008) and Large eddy simulation (LES) (Kurppa et al., 2018; Li et al., 2008; Sada and Sato, 2002). The choice among the three methods depends on the costs and objectives. One of the most important issues using CFD technology in the environment problems on the street-level is to obtain accurate initial and boundary conditions (Ehrhard et al., 2000). To solve this problem, the multi-scale coupling method is revealed as a good solution, which uses the meteorological information from mesoscale model as the initial and boundary conditions to drive CFD (Nelson et al., 2016). Tewari et al. (2010) proved that the CFD simulation was improved significantly when the results of Weather Research and Forecasting (WRF) model were used as the initial and boundary conditions. With the WRF model, the community multiscale air quality (CMAQ) model, and the CFD (RANS) approach, Kwak et al. (2015) built an urban air quality modelling system, which presented a better performance than the WRF-CMAQ model in simulating $NO_2$ and $O_3$ concentrations.

Nevertheless, the street-level air pollutant transport resulted from the nearby sources was still not fully investigated. The Flexible Particle (FLEXPART) model (Stohl et al., 2005; Stohl, 2003) is a gas-block trajectory-particle dispersion model based on the Lagrangian particle method. The FLEXPART model can track the transport of tracers via forward or backward simulation. Different from Eulerian model, the Lagrangian model is

not restricted by the Courant–Friedrichs–Lewy (CFL) condition (Stam, 1999) and thus, the integration process in the Lagrangian model can be maintained with high spatial resolution with acceptable computation efficiency. Initially, the FLEXPART model was driven by global meteorological reanalysis data from ECMWF or NCEP. Fast and Easter (2006) developed a FLEXPART version that used the WRF model output and was optimized with technical level and output results. Nowadays, the WRF-FLEXPART model has been widely used to research the regional transport of air pollutants (Yu et al., 2020; He et al., 2020; Gao et al., 2020; He et al., 2017a; Brioude et al., 2013; De Foy et al., 2011). Cécé et al. (2016) firstly applied the FLEXPART model at a small-scale resolution to analyse potential sources of $NO_X$ in urban areas, with the WRF-LES model results as the driving field. Though FLEXPART has been extensively applied in medium and long-range transport cases (Madala et al., 2015; Heo et al., 2015; Sandeepan et al., 2013; Liu et al., 2013), it has been rarely tested for street-level transport and small-scale resolution grids.

The objective of the present work is to investigate the flow field characteristics and potential impact of traffic sources to receptor sites, under real building scenarios and meteorological conditions. To this end, a multi-model simulation system for street level circulation and pollutant tracking (S-TRACK) was developed by integrating the WRF mesoscale, the STAR-CCM+ street scale and the FLEXPART particle dispersion models, and applied to the Jinshui District of Zhengzhou city, Henan Province. Zhengzhou is located in the central of China with four distinct seasons. According to the Oceanic Niño Index (ONI), an El Niño event occurred in January 2019. The occurrence of El Niño generally favours a warm winter and weak winter winds in China that is conducive to occurrence of air pollution. Therefore, the period of January 2019 was selected and

simulated for this study. This manuscript is organized as follows. Section 2 presents the model details and the observed data for the model validation. Section 3 provides the details of the model validation results, the wind environment characteristics and the potential impact of traffic source on receptor sites in the region. Section 4 provides the conclusions of the study.

## 2. Data and Methods

### 2.1 S-TRACK description

The S-TRACK system consists of three major components (Fig. 1). The WRF model is used to obtain the mesoscale three dimensions (3D) meteorological fields, with the initial and boundary conditions provided by NCEP FNL reanalysis data. The STAR-CCM+, driven by the meteorological data from WRF, is used to compute the refined 3D street-level meteorological fields with a resolution of 1 m to 100 m in the simulation area. With the refined 3D meteorology, the FLEXPART model is run to analyse the transports of traffic sources at street-level and their potential contribution to specific sites. One should note that some meteorological variables needed by FLEXPART that the STAR-CCM+ cannot provide (Table 1) are obtained from WRF simulations. The specific coupling scheme of the S-TRACK system is detailed as follows:

I. **Run the WRF model** (refer to Section 2.2 for specific settings) to obtain meteorological data with a spatial resolution of 1 km × 1 km, including temperature, pressure, humidity, wind, etc.

II. **Extract the value of temperature (T) and wind (U, V and W)** from the WRF simulation, as the initial and boundary conditions of the STAR-CCM+ simulation. Run the

STAR-CCM+ (refer to Section 2.3 for specific settings) to obtain values of meteorological variables with a spatial resolution of 1 m -  100 m, including wind field, surface pressure, and surface sensible heat flux, etc. The 3D street-level grid for STAR-CCM+ is detailed in Section 2.3.1.

III. **Match the STAR-CCM+ grids to the WRF grids.** As the FLEXPART-WRF (version 3.3.2) was used here, the grid structure of meteorological input data to FLEXPART should match the grid structure of WRF model. To this end, a regular fine grid with a horizontal resolution of 10×10 m was constructed base on the pre-processing system of WRF model (WPS). The urban building height data obtained base on drone aerial photography was taken as part of the terrain height data in the WPS. Once the refine grid was established, meteorological variables of the STAR-CCM+ and WRF model were interpolated into the grid by a nearest-neighbour interpolation method.

IV. **Run the backward FLEXPART model** (refer to Section 2.4 for specific settings) to obtain the 3D spatial location data of released particles, which was used to analyse the features of pollutant transport at street-level and potential contribution of traffic source to specific sites.

## 2.2 WRF model configuration

In this study, the WRF model is configured with four nested domains (Fig. 2a), with the resolution of 27 km × 27 km (85 × 85 grids), 9 km × 9 km (82 × 82 grids), 3 km × 3 km (82 × 82 grids), and 1 km × 1 km (61 × 61 grids), respectively. Vertically, there are 45 full eta levels from the surface to 100 hPa with 11 levels below 2 km, on which the meteorological fields are used to drive the STAR-CCM+. The innermost nested region is

shown in Fig. 2b, where the area focused in this study is marked with a black box. The

140 initial and boundary conditions of WRF model are obtained from the NCEP re-analysis

data (http://rda.ucar.edu/dataset-s/ds083.2). The boundary conditions are updated every 6

hr. Table 2 lists the selected physical parameterization schemes. The time from 12:00

Beijing time (BJT) on December 30, 2018 to 23:00 BJT on January 31, 2019 is chosen as

the modelling period, with the simulation results recorded every hour.

## 2.3 STAR-CCM+ configuration

The STAR-CCM+, one of the most commonly used commercial CFD software, was

selected for the street-level simulation. Previous studies had found an excellent correlation

between STAR-CCM+ simulated and measured values in simulating environmental and

meteorological problems at street-level (Santiago et al., 2017; Borge et al., 2018; Jls et al.,

2020). The model has functions such as geometric modelling, model pre-processing, the

calculation execution, and post-processing of results. More details on STAR-CCM+ can

be                                                    found                                                    at

https://www.plm.automation.siemens.com/global/zh/products/simcenter/STAR-

CCM.html.

## 2.3.1 3D street-level grid generation

The establishment of a 3D geometric model is based on the actual terrain and

buildings height data for the simulated area obtained through the drone aerial photography

technology. The basic data such as the geometric shape of urban buildings, roof height and

vector data of the top of buildings with high resolution, high timeliness and accuracy are

160 used to construct a realistic 3D geometric model for driving the STAR-CCM+ simulation.

In the process of model construction, the same shape as the actual building was maintained to reduce the influence of model errors on the calculation results (Fig. 3a). The length, width, and height of the STAR-CCM+ calculation domain are 13 km, 11 km, and 2 km, respectively, among which, nearly 2/3 of the buildings are distributed in the range of 10 - 40 meters, with the average height of the buildings of 32 m. The highest building in the area is 390 m, and the lowest building is 6 m.

The geometric model domain is divided by polyhedral meshes (Figs. 3c). The polyhedral mesh has much fewer cells than the traditional tetrahedral mesh, but with a similar accuracy of calculation. Under the same number of grid cells, the numerical simulation results of polyhedral grids are more consistent with experimental data than tetrahedral grid cells (Zhang et al., 2020). The grid cells on the ground and near the buildings are much denser (Fig. 3b) (the minimum resolution is about 1 m), so that the influence of the building on the flow patterns can be described more accurately. In the end, the number of unit grid cells generated is 382181, and the number of nodes is 1990224.

## 2.3.2 Physical model and boundary conditions

The STAR-CCM+ solves the RANS with the realizable k-ε turbulence closure scheme in this study (Li et al., 2019; Li et al., 2006; Lei et al., 2004). The ground and building surfaces are set to be no-slip, and the distribution of fluid velocity and pressure near the ground and the building surface is described by the blended wall function. For the coupling of WRF model to STAR-CCM+, the values of temperature and wind from the WRF simulation are extracted to establish the initial and boundary conditions for STAR-CCM+. Since the variables obtained by WRF simulation have a relatively coarse resolution

of 1 km, the velocity components (U, V and W) and the temperature are interpolated to the boundary of STAR-CCM+ domain using the spline interpolation method and the linear interpolation method, respectively. For the turbulence intensity and turbulence viscosity ratio, the lateral and upper boundaries are set as constants with values of 0.1 and 10, respectively.

## 2.4 FLEXPART configuration

The simulation area is set to sub-domain B in Figure 3, with a horizontal grid resolution of 10 m × 10 m. The simulation time is from 1:00 BJT 1 January 2019 to 23:00 BJT 30 January 2019. The time step of FLEXPART is 1 s, and the output time interval is 120 s. FLEXPART calculates particle trajectories using analysed winds plus random motions in order to account for turbulence. Simulation results from mesoscale meteorological models (such as WRF) do not resolve individual turbulence cells, although they reproduce the large-scale effects of turbulence. To account for sub-grid turbulence, turbulence options need to turn on in FLEXPART (Stohl and James, 2004). However in this study, the turbulence options are turned off, since the turbulence is already resolved by the STAR-CCM+ simulation. Through backward trajectory simulation, the impact of traffic source on the receptor sites in the region can be effectively analysed. Due to the high number of grids in the region and the fact that increasing the number of released particles leads to consuming more computational resources, the particle residence time is set as 2 h, and 5 tracer particles are released per hour, and the total number of particles released was 3590 tracer particles in the course of simulation.

## 2.5 Meteorological observation data

Hourly near-surface meteorological observations from the Bank School City monitoring site (hereinafter referred as the BSC monitoring site), including 2 m temperature (T), 2 m relative humidity (RH), surface pressure (P), 10 m wind direction (WD) and 10 m wind speed (WS) in January 2019 are used to evaluate the WRF and STAR-CCM+ simulation results, with the statistical indexes including Pearson's correlation coefficient (R), root mean square error (RMSE), mean bias (MB) and mean error (ME). The location of the BSC monitoring site (34.802375N, 113.675237E) is shown in Figure 3.

## 3. Results and discussions

## 3.1 Model evaluation

The performance of WRF model to simulate meteorological elements is an important basis for STAR-CCM+ and FLEXPART simulations. The hourly meteorological data for January 2019 obtained from the innermost nested simulation of the WRF model is selected to compare with observation data to verify the WRF model. Table 3 lists the statistical results of T, RH, P, and WS. The T and RH are slightly underestimated, with the MB values as 1.86 k and 5.95%, respectively, and the P and WS are overestimated by the WRF model, with the MB values as 3.66 hPa and 1.44 m s$^{-1}$, respectively. The R values for T, RH and P are 0.80, 0.70 and 0.98, respectively, passing the 99% significance test, and indicating that the variation characteristics of T, RH and P are well reproduced by the WRF model. WS is generally overestimated by WRF model (Temimi et al., 2020; He et al.,

2014), which is also found in the present study with the RMSE of 1.97 m s$^{-1}$. The performance of the near-surface meteorology obtained by the WRF simulation is equivalent to previous studies (He et al., 2017b; Carvalho et al., 2012).

Since the time-varying boundary conditions in the calculation domain of STAR-CCM+ are obtained from WRF model, the simulation performance of WRF model has an important influence on the STAR-CCM+ simulation results. The wind has an important influence on the transport of air pollutants in the area (Zhang et al., 2015). Figure 4 shows the hourly wind observations and simulations at the BSC monitoring site in January 2019. Both WRF and STAR-CCM+ overestimate the wind speed to certain degrees (Fig. 4a). The average of observed wind speed is 0.92 m s$^{-1}$, with the value simulated by WRF and by STAR-CCM+ is 2.37 m s$^{-1}$ and 2.00 m s$^{-1}$, respectively. The R values of WRF and STAR-CCM+ are 0.45 and 0.67, respectively, passing the 99% significance test, and demonstrating the refined STAR-CCM+ wind simulations are superior to that of the WRF. This might be due to the fact that the resolution of WRF simulation is not fine enough and the underlying surface is processed in a parameterized way that can't accurately describe the urban surface roughness. For the STAR-CCM+, the geometric model is used for the underlying surface, which could better reflect the urban surface conditions compared to parametric methods. Figure 4b shows the comparison results of the observed and simulated wind directions. It can be seen that the change of the wind direction is captured by the STAR-CCM+ well. The wind direction is verified by hit rates (HR) (Schlünzen and Sokhi, 2008). With desired accuracy between ±45∘, the HR are calculated at 63 and 51 % for STAR-CCM+ and WRF, respectively, indicating that variations in wind direction have been basically captured with a better performance for STAR-CCM+ simulations.

## 3.2 The characteristics of the street-level wind fields

In urban areas, the complex spatial structure and layout of buildings have a great influence on the street-level wind field (Liu et al., 2018; Park et al., 2015), which is a crucial meteorological factor that controls the transport of air pollutants. The street-level wind field characteristics were simulated by the S-TRACK and discussed comprehensively in this paper for the overall average in January as well as for different background wind directions, i.e., north, south, west and east, respectively.

### 3.2.1 The average wind field characteristics

Figures 5a-b illustrate the distribution of the average wind streamlines in January at the height of 5 m and 40 m, respectively. At the height of 5 m, the wind field structure is more complicated (Fig. 5a) than that at 40 m (Fig. 5b). The wind speed is relatively more intense in the areas where the buildings are sparse and smaller. In addition, the flow fields diverge or converge due to the layout of buildings and streets, causing the wind direction inside blocks differ from the background wind direction greatly. As the density of buildings gradually decreases with the increases of height, this phenomenon diminishes, reflected by the relatively more consistent wind fields at 40 m (Fig. 5b). The phenomenon was also found in a previous study (Sui et al., 2016).

To clearly show the details of the wind field, a sub-domain A (Fig. 3a) with complex building structures is selected from the entire computational domain. The near-surface winds disperse or converge horizontally and rise or subsidence vertically with the building (Fig. 5c). During the climb or fall with the building, downwash winds with high wind speeds occur (as shown in the red dashed circles). Due to the complexity of the building

layout, local circulation is formed on the west side of the BSC monitoring site, making the airflow around the building on the south side of the station accumulate and forms an obvious convergence area (Fig. 5c), which is not conducive to the air circulation and pollution transport (as shown in the red box).

**3.2.2 The wind field characteristics under different background wind directions**

Figure 6 shows the distributions of near-surface wind and its divergence under four different background wind directions. In general, the overall wind direction in the area is consistent with the background wind direction, but the airflow near-surface is significantly affected by the building layout, thus forming local circulations with divergence or convergence zones. The wind speeds in the areas with dense buildings are significantly smaller than those in open areas (Figs. 6a-1, 6b-1, 6c-1, and 6d-1), which is attributed to the obvious frictional dragging effect of the dense buildings. The overall wind direction in the area is generally the same as the background wind direction, but the airflow is diverged or converged by the influence of the building layout, resulting in a great difference in wind direction inside the block from the background. When the background wind direction is north or west (Figs. 6b and 6c), the overall wind speed in the area is relatively large. This is mainly due to the temperate monsoon climate in Zhengzhou, where northwest and west winds prevail in winter and wind speeds are relatively high.

It is found that the windward side of the building is mostly a divergence zone and the leeward side is mainly a convergence zone, which is more obvious for higher buildings. When the airflow meets the building, the airflow on the windward side of the building is blocked and thus spreads outward, forming a divergence zone; while the airflow on the

leeward side of the building converges and generates a vortex with lower wind speed, forming a convergence zone. For example, at BSC monitoring site, when the background wind direction is west, the wind speed on the windward side of the building is higher and diffused outward by the building blockage (Fig. 6c-2), resulting in a significant divergence zone (Fig. 6c-3). High-rise buildings have a greater impact on the wind field and cause a strong degree of convergence and divergence. It can be seen that the degree of divergence or convergence around the high-rise building is more significant than those around low buildings in the area (Figs. 6b-3, 6c-3, and 6d-3). In addition, the ventilation is better when the dominant airflow is in the same direction as building layout (Fig. 6c). In the process of urban construction, the influence of prevailing wind direction on the layout of buildings should be considered, which could effectively improve the efficiency of urban ventilation.

## 3.3 Potential impact of traffic sources

In this section, the S-TRACK system is used to analyse potential impact of main traffic roads (R1-R4) in sub-domain B (Fig. 3a) on several receptor sites nearby with different heights and locations with a number of schools and residential areas. The widths of roads R1-R4 are about 45, 33, 20 and 18 meters, respectively. During January 2019, the average traffic volumes were about 2300 (R1), 490 (R2), 400 (R3) and 90 (R4) cars per hour, respectively. Since the detailed information on road traffic emissions was not available, the road traffic emissions were assumed to be uniformly distributed and with identical intensity in this study. During the backward trajectory simulation, the particles as long as passing within 5 m height above the road is considered to be a potential contribution from the road emissions to the receptor site. Additionally, potential impact of

traffic source under different background wind directions was also explored. The residence-time analysis (RTA), which has been previously used to identify the accounted contribution of emission sources to air quality of receptors (Yu, 2017; Salvador et al., 2008; Hopke et al., 2005; Poirot et al., 2001; Ashbaugh et al., 1985), was selected in this study to assess the potential contribution ratio of the traffic source on receptors. The RTA is expressed as:

$$R_{i,j} = \frac{\tau_{i,j}}{t},$$

where $R_{i,j}$ indicates the contribution ratio of the grid (i,j) to receptor; $\tau_{i,j}$ means the residence time in the grid (i,j) and $t$ means the total residence time in all grids.

### 3.3.1 Potential impact of traffic source at different sites in winter

In order to analyse the potential impact of the traffic source on different locations, the receptor sites were selected at different locations and heights, and the overall potential contribution ratio of all wind directions for January 2019 was calculated by RTA (Table 4). Receptor sites S2-S8, with identical horizontal location but different heights, are selected to investigate contributions of traffic source to receptor sites at different heights. The potential contribution ratios of all the four roads are 4.05%, 4.25%, 4.33%, and 4.67% for receptor sites S2 to S5, with the height of 2 m, 5 m, 10 m, and 15 m, respectively. However, as the receptor height continues to rise, namely from S5 to S8, the potential contribution ratios of the roads gradually decrease from 4.67% to 3.55% (Table 4). It's noteworthy that the contributions from R1 and R3 are primary, especially the R1, which may be due to the closer distance to the site and the generally northeast wind field. The

potential impact of the traffic source is the greatest when the receptor site is located at a height of 15 m, suggesting the air quality at that height is most susceptible to traffic emissions under the northeast wind field. In addition, according to density distribution of all trajectory points that have passed through the traffic roads (Figs. 7), it can be seen that the road section with large potential impact to the receptor sites generally located to their northeast, which might be a result of the combination effect of the background wind field and the building layout. For more details, the vertical structure of winds along the direction of the wind field at the receptor site S2 (Fig. 8b) is also presented. It can be seen that there is a general upward motion in the airflow, making the position with the greatest potential contribution ratio from traffic source locate at a certain height, which is about 15 m over the receptor site S2 in this case.

It can be seen from Table 4 that R1 is the road with the greatest potential impact to the receptor sites. The horizontal distance between road R1 and the receptor sites is about 300 m and the peak of the potential contribution ratios occurs at a height of about 15 m (corresponding to the site S5). However, for the road R3, which is closest to the receptor sites in horizontal (about 200 m), the contribution ratios are lower than those of the road R1. Figure 8a shows that the near ground winds are generally northeast, resulting in that the probability of traffic contributions from R1 and R3 road sections upwind of the site S2 is roughly the same. Nonetheless, as mentioned in section 3.3, the width for the road R1 is about twice of that for the road R3. Therefore, even R1 was a little farther from the receptor sites than R3, the contribution ratios of R1 to the sites were calculated larger than those of R3. For the R2 and R4, the distance from the receptor sites is about 1200 and 1500 m, respectively, far away than those of R1 and R3. In addition, under the northeast winds,

traffic source was hardly transported to the receptor sites, rendering the contribution ratios quite small below 50 m (Table 4). It can also be seen, from Table 4, that the corresponding potential contribution ratios of R2 and R4 may peak at a height over 50 m.

Since the road R1 had the largest potential contribution to the receptor sites, the contribution of R1 to different positions is focused in the subsequent discussion. For the receptor site S1, which is about 400 m from the R1, located in a dense building area with the building height at 30 to 40 meters, the potential contribution ratio of the traffic source on the receptor site is calculated to be 1.81%. For the receptor site S2, which is about 300 m from the traffic road, located in an open area and surrounded by low buildings, the potential contribution ratio of the traffic source is determined to be 2.38%. It might be inferred that the wind field difference partially resulted from the influence of buildings layout led to the higher contribution ratio to S2. From the average wind field in January 2019 (Fig.8a), it can be seen that the winds were influenced by high-rise buildings around the S1, resulting in a change in transport path of pollutants and thus, making pollutants difficult to reach the S1 site. However, for S2 site, the winds were less influenced by the buildings and pollutants were more easily transported there.

**3.3.2 Potential impact of traffic source under different background wind directions**

In order to investigate the potential impact of traffic source under different background wind directions, the receptor site S2 influenced by the R1 under the east, the south, the west, and the north wind directions was classified from the simulation period. The potential contribution ratios of traffic source were estimated to be 2.45%, 0.07%, 1.98%, and 2.97% for the east, the south, the west, and the north wind directions,

respectively, revealing that the difference in potential impact was largest between the south and north wind directions. When the background wind direction was south, the receptor site was located upwind of the road, and the road traffic source contributed very little to the receptor site. On the contrary, when the receptor site was downwind of the road with northern winds, the contribution ratio of road traffic source to the receptor site was the

greatest. When the background wind direction was east and west, the contribution ratio to the receptor point was similar, ranging between the ratios under south and north wind directions. The lower contribution ratio during westerly winds relatively to that under easterly winds might partially be due to the denser distribution of buildings upwind of the receptor site. Complex building layouts changed the structure of the wind field and thus

had an impact on the transport of pollutants. The slow air circulation in dense building areas made it unfavourable for pollutants to be transported. In the windward side of the dense building area, the wind was blocked and diverted to both sides of the building. Pollutants were difficult to transport to the leeward side of the building, where the receptor site was located. The results of the potential impact of traffic source under different

background wind conditions is helpful to understand the streel-level pollution transport characteristics and provides effective suggestions for the traffic pollution control strategies.

## 4. Conclusions

A street-level pollutant tracking system has been developed to simulate micro-scale meteorology and used to analyse the characteristics of wind environment and the potential

traffic source contribution of air pollution to receptors through backward simulations in a city district. In general, the S-TRACK system is effective in simulating the street-level

meteorological and pollution problems. The presence of buildings has a significant effect on the wind environment, i.e., the dragging effect of dense buildings renders the wind speed inside the block smaller than the background wind speed. The ventilation is better when the dominant airflow is consistent with the direction of building layout. Influenced by the building layout, the airflow near-surface is formed with divergence and convergence zones. The windward side of the building is mostly a divergence zone and the leeward side is mostly a convergence zone, which is more obvious for higher buildings.

As a test case, the S-TRACK system has been used to investigate the potential impact of traffic source on receptor sites with different locations, heights and background wind directions in a city district. The potential impact of traffic sources on a specific receptor site varies under different background wind directions, which are estimated to be 2.45%, 0.07%, 1.98%, and 2.97% for the east, the south, the west, and the north wind directions, respectively. The difference in potential contribution under east and west wind directions might partially be due to the density of buildings upwind of the receptor site. For a specific location of this case study, the potential traffic contribution ratios also varied with height at about 4.05%, 4.25%, 4.33%, 4.67%, 4.38%, 3.64% and 3.55% for 2, 5, 10, 15, 20, 40, 50 m, respectively, manifesting a significant trend of increasing and then decreasing with height. In addition, the height of position with the greatest potential contribution ratio from the traffic source varies jointly influenced by the distance between the position and traffic source, as well as the background wind field.

In the future, in-depth simulation experiments with different building layouts, wind field environments, and distances between traffic source and receptor are required to quantify the potential impact of street-level pollution sources  and to establish the

relationship between meteorological conditions, buildings and various emissions (point, area and line sources) in the street-level for an effective management of regional pollution in a city.

## Appendix

## 1. Some settings to improve the calculations efficiency of CFD

It is true that using a CFD model for the atmospheric numerical simulation has the problem of high computational cost. In this study, the RANS is chosen as the CFD preprocessing model, which requires relatively small amount of computational resources. The time step of STAR-CCM+ is set to 60s, with a maximum of 20 internal iterations in each time step and a parallel computing with 32 CPUs is done on a supercomputer. The simulation error increases with the simulation time. In order to ensure the efficiency and accuracy of the simulation, the month was divided into four time periods to simulate, as shown in Table A1.

**Table A1. The division of each simulation time period and the physical time spent on the simulation**

| Simulation start time | Simulation end time | Length of simulation time | Physical time spent |
|---|---|---|---|
| 2018/12/31 00:00:00 | 2019/1/ 09 04:00:00 | 220h | 126.45h |
| 2019/1/08 00:00:00 | 2019/1/ 17 04:00:00 | 220h | 128.33h |
| 2019/1/ 16 00:00:00 | 2019/1/25 04:00:00 | 220h | 128.53h |
| 2019/1/24 00:00:00 | 2019/2/01 08:00:00 | 200h | 117.10h |

## 2. Significance test

Significance test is used to determine the significance of the results in relation to the null hypothesis, with a p-value, or probability value describing how likely the data would

have occurred by random chance (i.e. that the null hypothesis is true). A p-value less than 0.05 (typically $\leqslant 0.05$) is statistically significant. It indicates strong evidence against the null hypothesis, as there is less than a 5% probability the null is correct.

## 3. The observed data for January 2019 at various meteorological stations in Zhengzhou city.

**Table A2. The location of each meteorological station and the average wind speed.**

| number | Latitude and longitude coordinates | Average wind speed |
|--------|-----------------------------------|--------------------|
| 1 | (34.7274 N, 113.7493 E) | 0.92 m s$^{-1}$ |
| 2 | (34.73506 N, 113.6457 E) | 0.92 m s$^{-1}$ |
| 3 | (34.7466 N, 113.6876 E) | 1.32 m s$^{-1}$ |
| 4 | (34.76117 N, 113.6883 E) | 0.61 m s$^{-1}$ |
| 5 | (34.78245 N, 113.6567 E) | 1.51 m s$^{-1}$ |
| 6 | (34.81151 N, 113.6948 E) | 1.48 m s$^{-1}$ |
| 7 | (34.83267 N,113.5453 E) | 0.72 m s$^{-1}$ |

## 4. Divergence

The divergence is a quantity that describes the degree to which air converges from its surroundings to a point or flows away from a point. It is used to describe the intensity of divergence and convergence at locations in space. The formula is as follows.

$$div\ \mathrm{v} = \nabla \bullet \mathrm{v} = \frac{\partial u_i}{\partial x_i} = \frac{\partial u}{\partial x} + \frac{\partial v}{\partial y} + \frac{\partial w}{\partial z},$$

where $u$, $v$ and $w$ are the components of the wind in the $x$, $y$ and $z$ direction, respectively. When the $div\ \mathrm{v} < 0$, the location is convergence; when the $div\ \mathrm{v} > 0$, the location is divergence.

## 5. PBLH validation

The bulk Richardson number (Ri) method was taken to estimate the BLH base on the

sounding data of Zhengzhou. Ri is expressed as:

$$R_i(z) = \frac{(g/\theta_{vs})(\theta_{vz} - \theta_{vs})(z - z_s)}{(u_z - u_s)^2 + (v_z - v_s)^2 + (bu_*^2)} \,,$$

where $z$ is the height above ground, $s$ the surface, $g$ means the acceleration of gravity, $\theta_v$

the virtual potential temperature, $u$ and $v$ the component of wind speed, and $u_*$ the surface

friction velocity. $u_*$ can be ignored here due to it is small relative to the wind shear

(Vogelezang and Holtslag, 1996). Previous theoretical and laboratory studies suggested

that when Ri is smaller than a critical value (~0.25), the laminar flow becomes unstable

(Stull, 1988). Therefore, the lowest level z at which the interpolated Ri exceeds the critical

value of 0.25 is referred to as PBLH in this study, which is referred to the criterion used

by Seidel et al. (2012). The R value is 0.57, passed the 99% significance test. It can be

seen from Figure A1 that the variation of boundary layer height is generally captured.

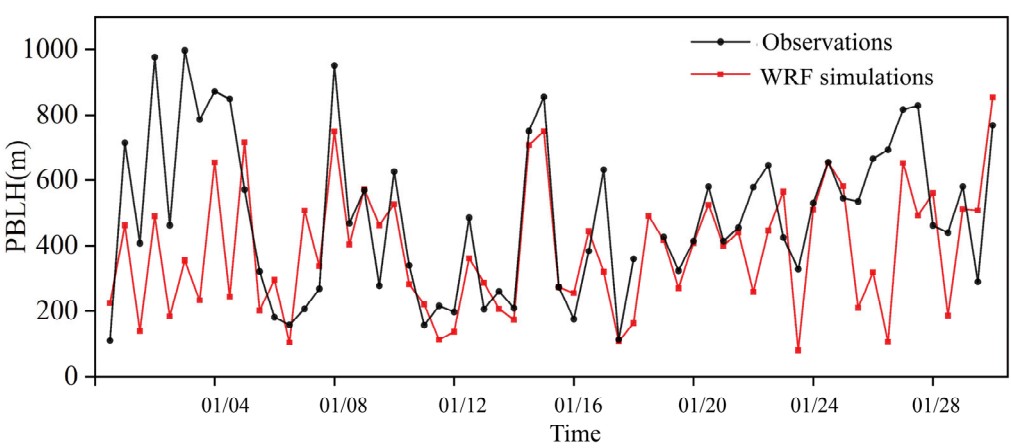

**Figure A1. Time series of the observed (black) and simulated (red) PBLH at 8:00 and 20:00 Beijing
time (BJT) in Zhengzhou sounding site.**

## Code/data availability

All source code and data can be accessed by contacting the corresponding authors Sunling Gong (gongsl@cma.gov.cn) and Lei Zhang (leiz09@cma.gov.cn).

## Authors contribution

Sunling Gong and Lei Zhang designed the research. Huan Zhang performed the simulations and wrote the manuscript with suggestions from all authors. Jingyue Mo, Huabing Ke, and Shuhua Lu assisted with data processing. Jianjun He, Yaqiang Wang and Lixin Shi participated in the scientific interpretation and discussion. All authors contributed to the discussion and improvement of the manuscript.

## Competing interests

The authors declare that they have no conflict of interest.

## Acknowledgments

The authors would like to acknowledge Bin Cui and Lin Zhang from Peking University and Liangfu Chen from Chinese Academy of Sciences for their valuable suggestions to improve the article.

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

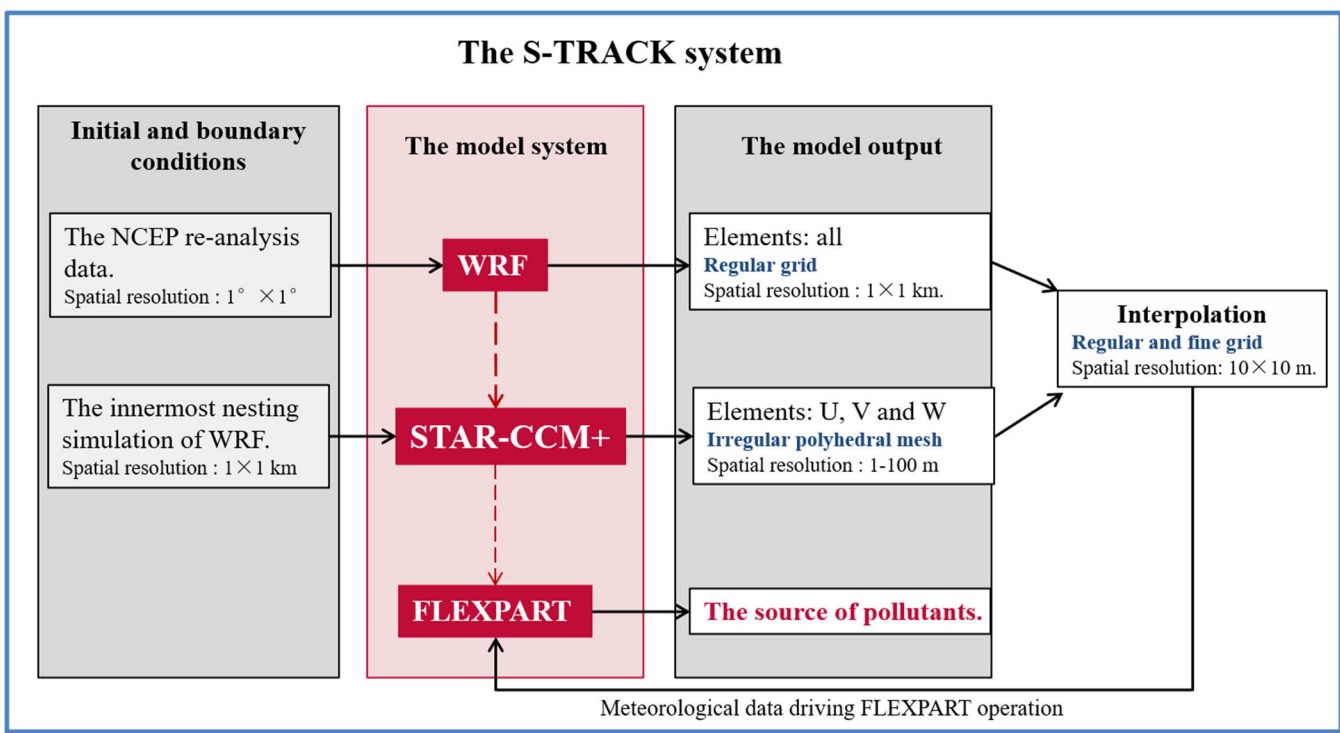

**Figure 1: The S-TRACK system: The role of WRF, STAR-CCM+ and FLEXPART in the S-TRACK system and the process of gradual refinement of resolution.**

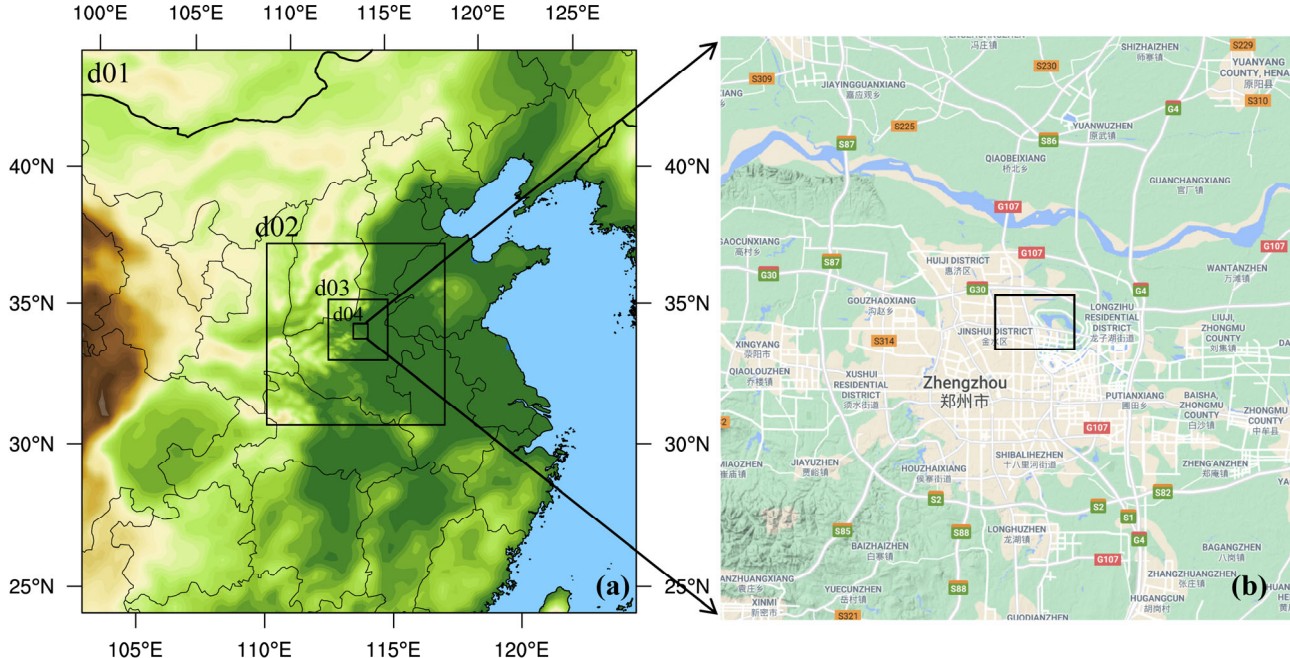

**Figure 2: Domain configuration of the WRF model: (a) the range of the four nested domains (d1-d4); (b) the innermost nested domain (d4), within which the black box represents the STAR-CCM+ simulation domain (extracted from © Google Maps 2021).**

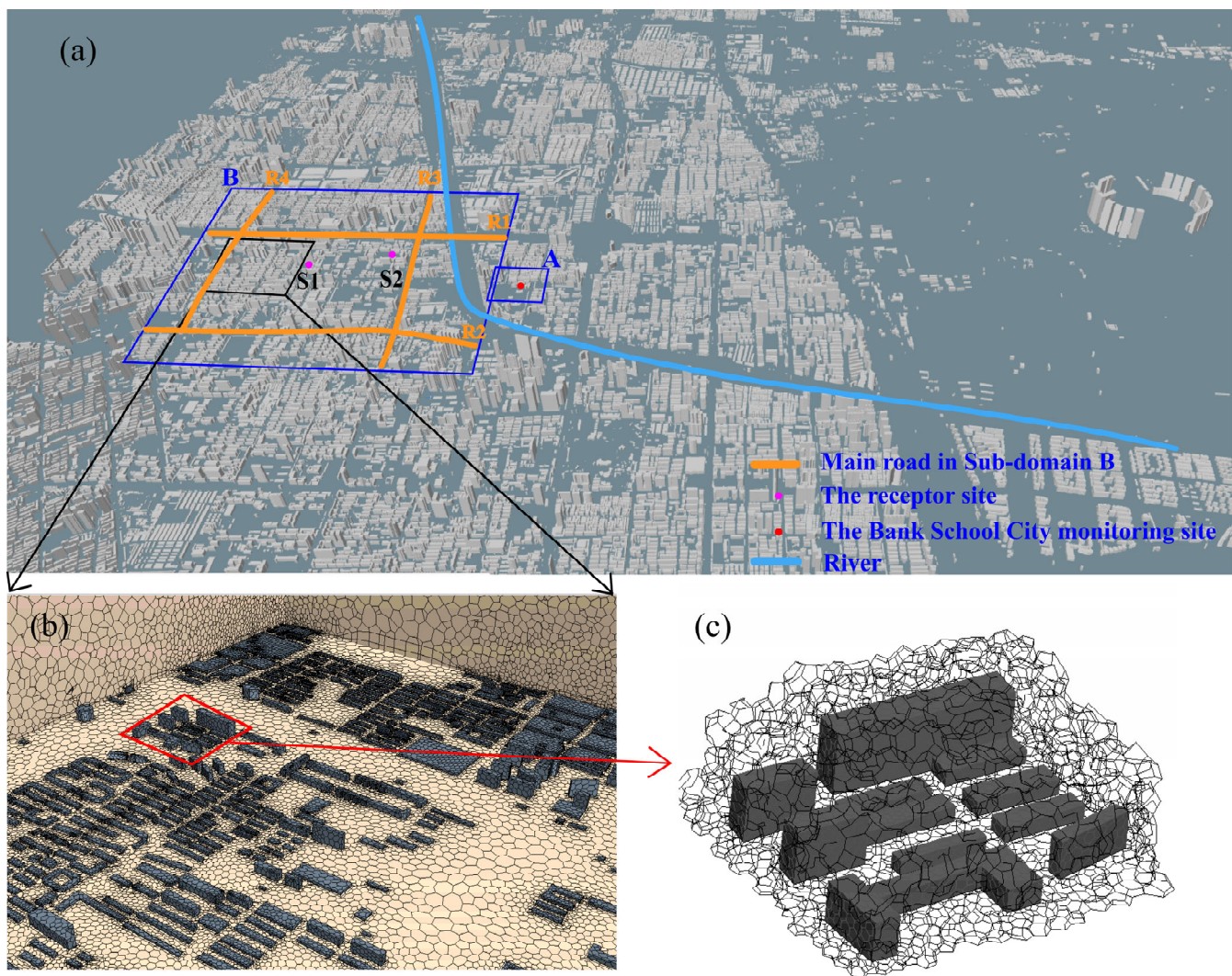

**Figure 3:** The computational domain of the STAR-CCM+ is shown in (a). The sub-domain A is used for detailed analysis of the wind environment, and the Bank School City (BSC) monitoring site is marked with the red dot. Sub-domain B is used to analyse the potential impact of traffic source on receptor sites in the region, with magenta dots (S1 and S2) indicating the receptor sites and Orange line indicating the main roads. The polyhedral mesh is used to divide the STAR-CCM+ simulation area. The mesh details of the vertical cross section and building surface are shown in (b), and the 3D meshes are shown in (c).

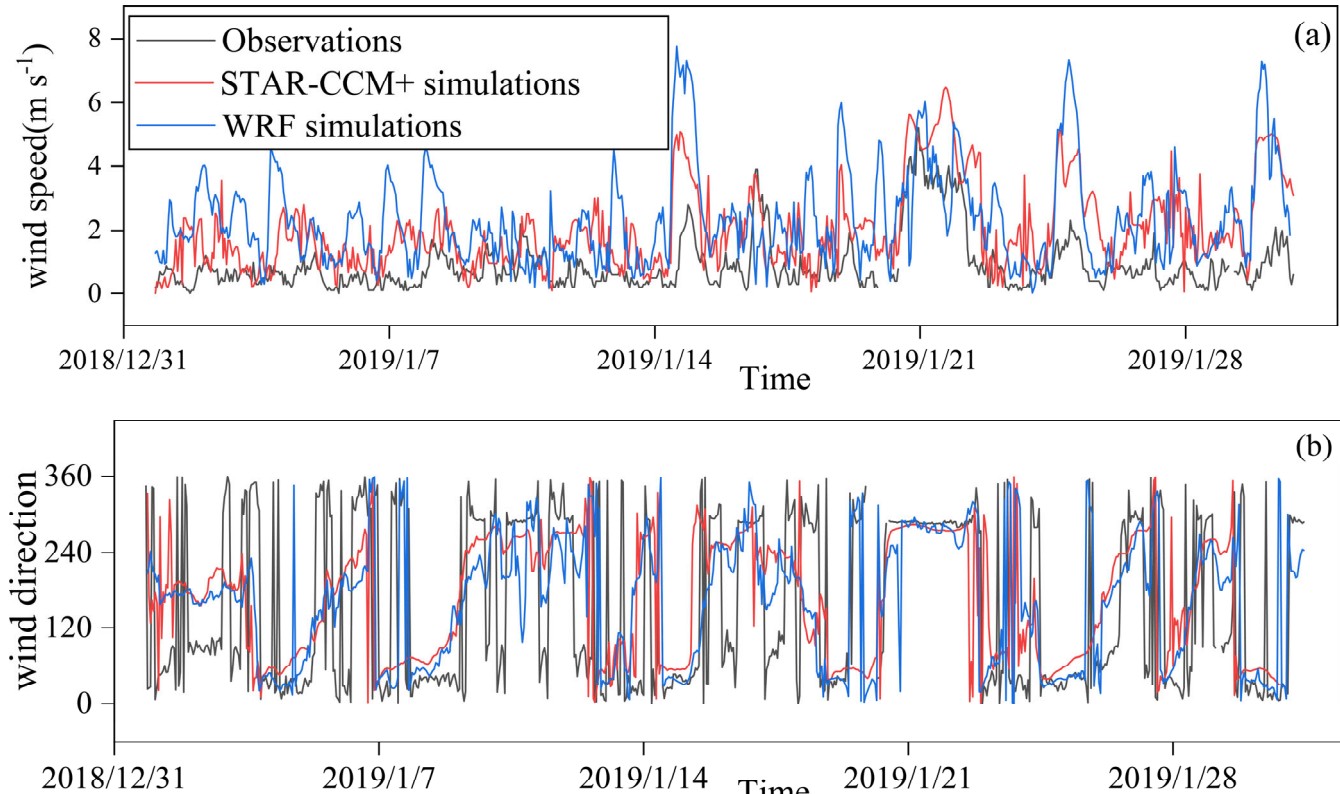

**Figure 4: Evaluation of the wind simulation results at the BSC monitoring site (see in Fig. 3a): the simulated, by WRF (blue line) and STAR-CCM+ (red line) model, respectively, and the observed (grey line) hourly near-surface wind speeds (a) and wind directions (b).**

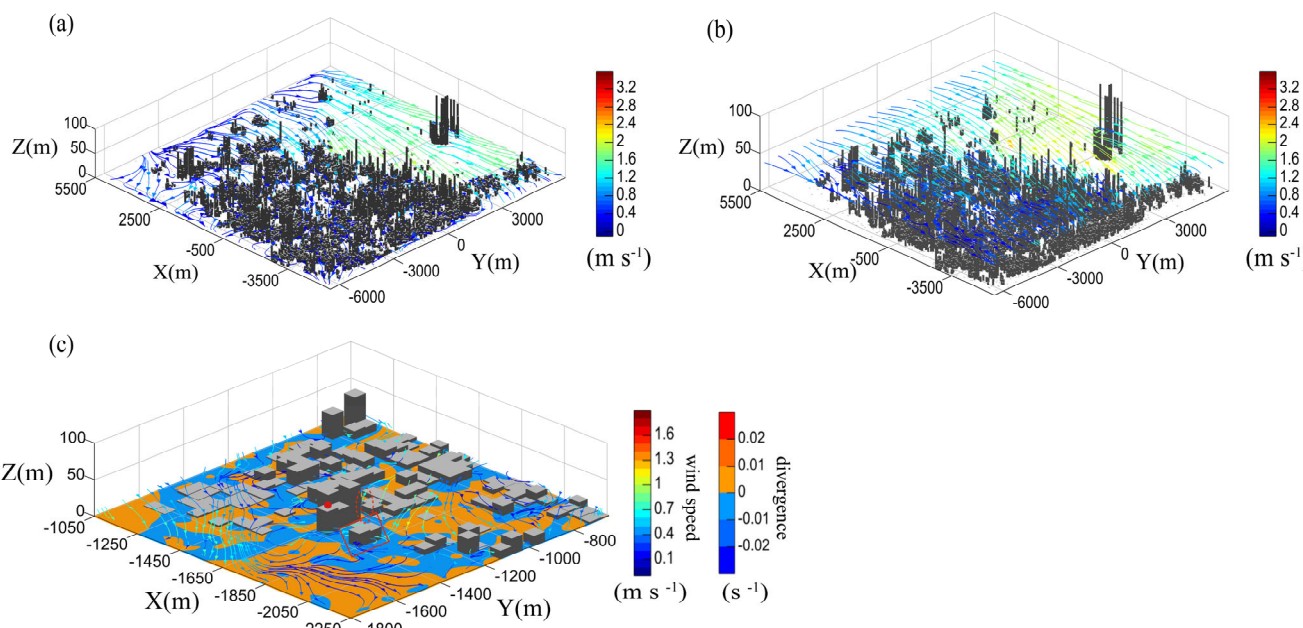

**Figure 5: The simulated wind streamlines at the height of 5 m (a) and 40 m (b) averaged in January 2019 in the whole S-TRACK simulation domain; the simulated wind streamlines and divergence (c) at the near-surface averaged in January 2019 in the sub-domain A (see in Fig. 3a). The BSC monitoring site is marked with red dot.**

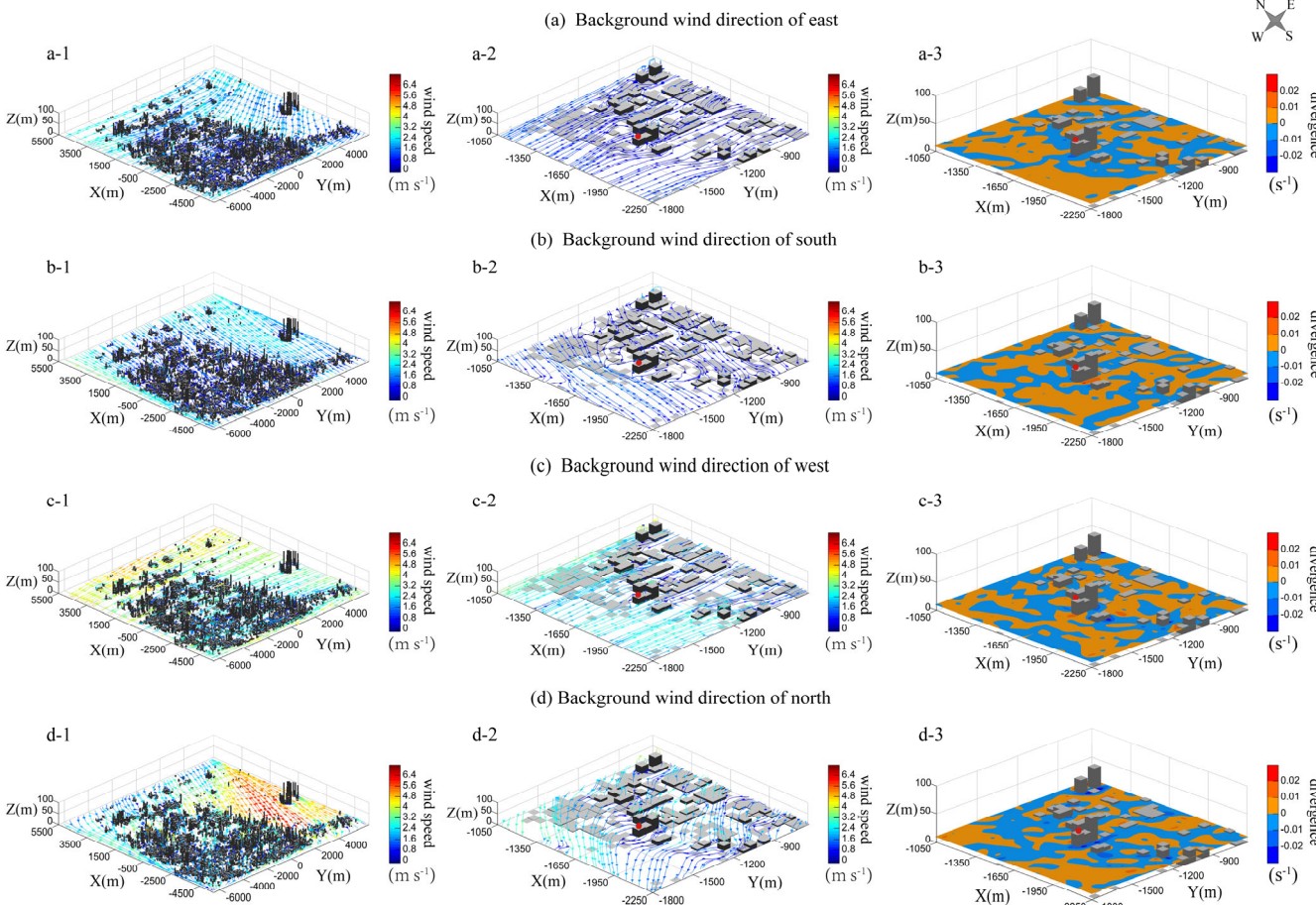

**Figure 6: The wind field streamlines and divergences under the background wind directions of east (a), south (b), west (c) and north (d). The BSC monitoring site is marked with red dot.**

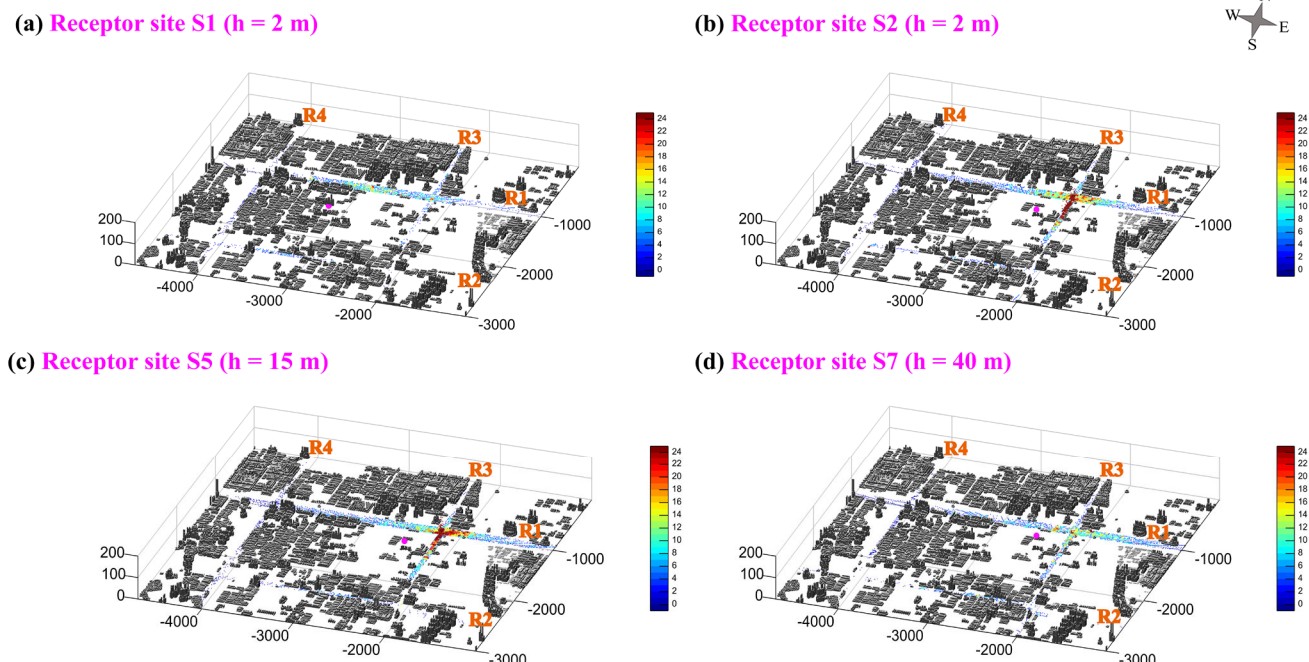

**(a)** Receptor site S1 (h = 2 m)   **(b)** Receptor site S2 (h = 2 m)

**(c)** Receptor site S5 (h = 15 m)   **(d)** Receptor site S7 (h = 40 m)

**Figure 7: Density distribution (refers to the number of particles that have stayed in the space of 10 m × 10 m in the horizontal direction and 5 m from the surface to above in the vertical direction) of all trajectory points passing through the traffic roads that received from different receptor sites (S1, S2, S5, and S7, see details in Table 4). The four receptor sites are all marked with magenta dots.**

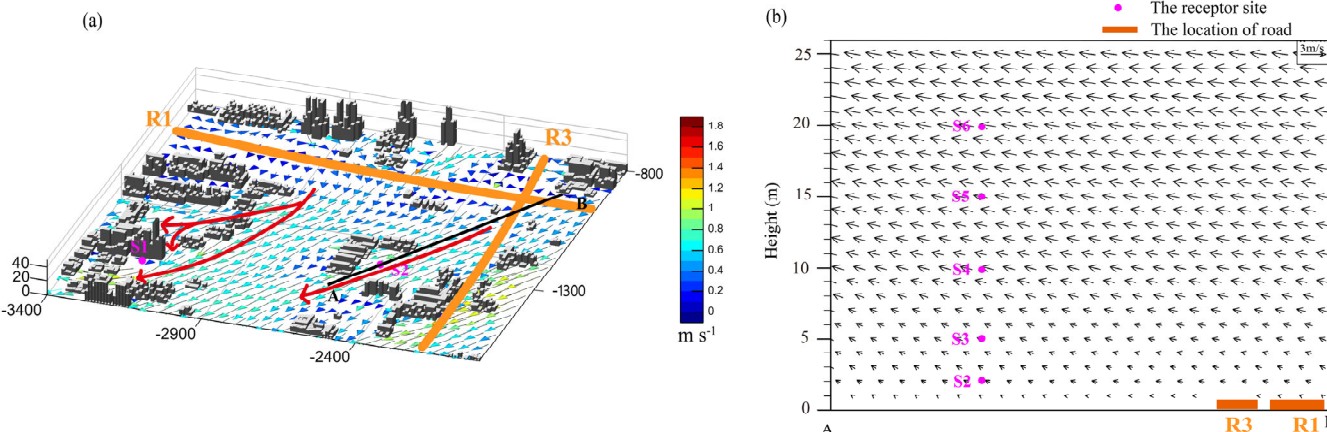

**Figure 8: The (a) average surface wind and (b) vertical structure of average winds that along the wind direction around the receptor site S2 (line AB) in January 2019. The road R1 is marked with orange line; the location of the vertical profile is shown in black line, and the receptor sites S1 to S6 are all marked with magenta dots.**

**Table 1. The list of variables required to run FLEXPART and the sources of variables.**

| variable | Description | Source |
|---|---|---|
| PB | base value of pressure | WRF |
| P | perturbation of pressure | WRF |
| PHB | base value of geopotential | WRF |
| PH | perturbation of geopotential | WRF |
| T | temperature | WRF |
| QVAPOR | specific humidity | WRF |
| MAPFAC_M | map factor | WRF |
| PSFC | surface pressure | STAR-CCM+ |
| U10 | 10 m wind along $x$ axis | STAR-CCM+ |
| V10 | 10 m wind along $y$ axis | STAR-CCM+ |
| T2 | 2 m temperature | WRF |
| Q2 | 2 m dew point | WRF |
| SWDOWN | surface solar radiation (optional) | WRF |
| RAINNC | large scale precipitation (optional) | WRF |
| RAINC | convective precipitation (optional) | WRF |
| HFX | surface sensible heat flux (optional) | STAR-CCM+ |
| U | wind along $x$ axis | STAR-CCM+ |
| V | wind along $y$ axis | STAR-CCM+ |
| W | Cartesian vertical velocity | STAR-CCM+ |

**Table 2 Parameterization scheme for the physical processes set up in the WRF model.**

| Physical management | Parameterization | Reference |
|---|---|---|
| Microphysics scheme | Lin | Lin et al. (1983) |
| Longwave radiation scheme | RRTMG | Iacono et al. (2008) |
| Shortwave radiation scheme | RRTMG | Iacono et al. (2008) |
| Land surface scheme | Noah | Chen and Dudhia (2001) |
| Planetary boundary layer scheme | MYNN3 | Nakanishi and Niino (2006) |

700

**Table 3 Statistical performances of the hourly near-surface meteorology simulated by the WRF model.**

| | R | MB | ME | RMSE |
|---|---|---|---|---|
| T | 0.80 | -1.86 (K) | 2.33 (K) | 2.82 (K) |
| RH | 0.70 | -5.95 (%) | 11.5 (%) | 15.0 (%) |
| P | 0.98 | 3.66 (hPa) | 3.66 (hPa) | 3.77 (hPa) |
| WS | 0.45 | 1.44 (m s$^{-1}$) | 1.58 (m s$^{-1}$) | 1.97 (m s$^{-1}$) |

**Table R4. Locations of receptor sites and the corresponding potential contribution ratios.**

| Receptor site | Location (x, y, z) | potential contribution ratio | | | | |
| --- | --- | --- | --- | --- | --- | --- |
| | | R1 | R2 | R3 | R4 | All |
| S1 | (-3200 m, -1420 m, 2 m) | 1.81% | - | - | - | - |
| S2 | (-2500 m, -1300 m, 2 m) | 2.38% | 0.18% | **1.32%** | 0.16% | 4.05% |
| S3 | (-2500 m, -1300 m, 5 m) | 2.57% | 0.29% | 1.28% | 0.10% | 4.25% |
| S4 | (-2500 m, -1300 m, 10 m) | 2.71% | 0.32% | 1.18% | 0.12% | 4.33% |
| S5 | (-2500 m, -1300 m, 15 m) | **2.98%** | 0.27% | 1.22% | 0.20% | **4.67%** |
| S6 | (-2500 m, -1300 m, 20 m) | 2.75% | 0.37% | 1.09% | 0.17% | 4.38% |
| S7 | (-2500 m, -1300 m, 40 m) | 2.30% | 0.39% | 0.70% | 0.25% | 3.64% |
| S8 | (-2500 m, -1300 m, 50 m) | 1.94% | 0.57% | 0.68% | 0.36% | 3.55% |

705