# Peer review of "Development and application of a street-level meteorology and pollutant tracking system (S-TRACK)"

_Atmospheric Chemistry and Physics, 2021_

## Author Comment (AC2)

Dear editor and referee#2:

Thank you very much for your time and attentions on this work. The comments and suggestions are very useful to improve our manuscript. Following is a point-by-point response to referee #2's comments. Texts in italic are the comments, and those in black bold are our responses. We hope that you will find the changes satisfactory.

The authors presented an interesting work that integrates the Weather Forecasting Model (WRF), Computational Fluid Dynamics (CFD) and Flexible Particle (FLEXPART) to study local circulations at a neighbourhood scale and the potential contribution of one street as a source of air pollution. The authors analyzed local wind circulation under different regional wind regimes. The methods used by the authors are robust and the meteorology simulated shows good agreement as shown in figure 4. However, conclusions, as stated in lines 338-339, authors say: "In general, the S-TRACK system is effective in simulating meteorological and air pollution problems" has no strong basis. This is because the is no real emissions and only one street was evaluated. Specifically:

**Specific comments**

1) in the abstract, the authors state: "The results of the study are helpful to understand the characteristics of wind environment and effect of traffic emissions in the area...", however, there are no traffic emissions in the study.

Response: Thank you sincerely for pointing it out. In the current study, we did not have the detailed information about the traffic emissions from the road, so we assumed that the road traffic emissions are uniformly distributed. During the backward trajectory simulation, the particles as long as passing within 5 m height above the road is considered to be a potential contribution from the road emissions to the receptor site. Using the methods of residence time analysis (RTA) to determine the magnitude of the potential contribution, the RTA is expressed as follows.

$$R_{i,j}=\frac{\tau_{i,j}}{t},$$

where  $R_{i,j}$  indicates the contribution ratio of the grid (i, j) to receptor;  $\tau_{i,j}$ means the residence time in the grid (i, j) and t means the total residence time in all grids. There is some uncertainty in this approach. Your suggestion is much appreciated, and we have modified the section as: "The results of the study are helpful to understand the characteristics of wind environment. With the hypothesis that traffic emissions are uniformly distributed on roads, the effect of traffic emissions in the area is revealed as well, which is important to improve urban living environment and control air pollution." In addition, we also provide additional clarification for this situation in Section 3.3.

2) The authors presented the potential impacts of only one street. however, we do not know anything about that street, type of traffic, the diurnal cycle of traffic by type of vehicle and as shown in Figures 3a, there are many streets surrounding the area. For instance, Figure 7b shows that the street might be important for site S2, but this site is closer to a perpendicular street (east), as shown in figure 3a. In a summary, despite that the results are interesting, more work is needed to provide the basis for the conclusions.

Response: Indeed, there are many roads around the area that may have impacts on the receptor sites. According to your suggestions, we have selected four main roads in the region as shown in revised Figure R4a to analyze the potential impact characteristics of traffic sources on receptor sites. The widths of roads (including greenery and open space) R1-R4 are about 90, 40, 30 and 30 meters, respectively. We have done our best to collect the information of the roads. During January 2019, the average hourly traffic volumes are about 2300 (R1), 490 (R2), 400 (R3) and 90 (R4) cars, respectively. We assume that the road emissions are uniformly emitted. The magnitude of the potential contribution of the four main roads of the sub-domain B to each receptor site is shown in the Table R4. Following your suggestions, we have analyzed the potential contribution of the four roads to the receptor sites (S2-S8) at different heights separately in Section 3.3.1 in the revised manuscript, and found that road R1 had the largest potential contribution to the receptor sites. Accordingly, in the subsequent analysis in the manuscript, we only considered the contribution of R1 to different positions (i.e., S1 and S2 sites).

---

## Author Response (AR1)

Dear Editor and referees,

We are very grateful for your time and attentions on this manuscript.

Please find below our itemized responses to the referees' comments.

We have addressed all the comments raised by both referees and

5 incorporated them in the revised manuscript.

Thank you for your consideration.

Sincerely,

Sunling Gong, Lei Zhang, Huan Zhang et al.
* * *
**Referee #1**

15    General comments

  *The authors present a new multi-modeling system S-TRACK which combines the Weather Research and Forecasting (WRF) model for the regional meteorological modeling and the STAR-CCM+ model for the street-scale air flow modeling and the FLEXPART model for a backward trajectory modeling.*

  *The multi-scale modeling is an important issue to improve modeling performance of street-scale meteorology and air quality. In this manuscript, downscaling is conducted from the linking between the regional-scale model to the street-scale model. However, more detailed description for the linking is necessary (see my specific comments 1 and 5).*

  *The most interesting part is a new research method to track the sources of pollutants in the street level using a backward trajectory modeling. The authors should give more analysis on the results as suggested in my specific comments 2 and 3.*

Specific comments

*1. The authors need to explain in more details how three components of S-TRACK are coupled. What is the major development by the authors to obtain the final simulation results? For example, in Figure 1, a regular and fine grid having the spatial resolution of 10 m is constructed by combining WRF and CFD results. Why is this refined grid necessary for FLEXPART? How is it done? An another example is in l. 164. According to the authors, WRF results fill some missing data from CFD results. It is not clear. What data are missing in CFD model? How have the authors filled this gap?*

  Response: **Thanks for pointing this out. The most important development we have made in this study was to integrate the WRF, STAR-CCM+ and FLEXPART models into a coupled system called S-TRACK. The main techniques include: facilitating the simulation results of the WRF model into STAR-CCM+ as the initial and boundary conditions, building a 3-D street model to be used by STAR-CCM+, constructing a high-resolution grid and interpolating the simulation results of the WRF and STARCCM+ model into the high-resolution grid to ensure the implementation of the FLEXPART model in the street-level.**

  **In the research of environmental meteorological problems at the street level, the grid resolution is usually set to several meters to tens of meters**

50 **(Kwak et al., 2015; Santiago et al., 2017). In view of the simulation duration, simulation range, computational accuracy and computational efficiency in this study, a regular fine grid with a spatial resolution of 10x10 m was constructed from the un-regular STARCCM+ grid to be coupled with FLEXPART (FLEXPART-WRF version 3.3.2). Therefore, the required**

55 **meteorological data format should all match the data formats by the WRF simulations. It was done in the following orders. Firstly, the horizontal grid was constructed with a grid resolution of 10m×10m and the building height was imported into the terrain data by using the pre-processing system of WRF model (WPS), and the vertical grid is constructed and divided into 32 layers**

60 **(up to 2 km) using WRF model.**

**Due to the limitations of the STAR-CCM+ model, some of the meteorological variables (as shown in Table R2) that are needed by FLEXPART cannot be produced and need to be provided by WRF. The WRF simulation domain is divided by a regular mesh but with coarse mesh**

65 **resolution, while the STAR-CCM+ simulation domain with a fine mesh resolution but is divided by an irregular polyhedral mesh. Therefore, it is necessary to interpolate the output of WRF mode and STAR-CCM+ mode into the refined grid constructed above to generate the meteorological input file that drives the FLEXPART model. The selected interpolation method in this**

70 **study is the nearest neighbor interpolation method. We have provided additional details on the specific coupling scheme of the S-TRACK model system in Section 2.1 of the manuscript.**

**Reference:**

75 Kwak, K. H., Baik, J. J., A, Y. H. R., and Lee, S. H.: Urban air quality simulation in a high-rise building area using a CFD model coupled with mesoscale meteorological and chemistry-transport models - ScienceDirect, Atmospheric Environ., 100, 167-177, http://dx.doi.org/10.1016/j.atmosenv.2014.10.059, 2015.

Santiago, J. L., Rafael Borge, Fernando Martín, David de la Paz, Alberto Martilli, Lumbreras, J., and Sá
80 nchez, a. B.: Evaluation of a CFD-based approach to estimate pollutant distribution within a real urban canopy by means of passive samplers., Sci. Total Environ., 576, 46-58,

https://doi.org/10.1016/j.scitotenv.2016.09.234, 2017.

85 *2. I think that one of the major findings in this manuscript is the potential impact of the traffic source is the largest at 15 m (l. 309). However, the authors' interpretation is not clear. They linked the result to the distance of the building from the source. This result should be explained in details taking into account the background wind field. And this result should be added in Conclusion.*

90 **Response: Following your suggestions, the wind field averaged in January 2019 is shown in Fig. R1a. It can be seen that the overall wind direction is northeast (Fig. R1a), which is consistent with the fact that the road section with lager potential impact to air pollution at receptor sites is located to their northeast. For more details, we also presented the vertical structure of winds that along the direction of the wind field at the receptor site S2 (Fig. R1b). It can be seen that there is a general upward trend in the airflow, making the position with the greatest potential contribution rate from traffic source located at a certain height, which is about 15 m over the receptor site S2 in this case. It can be inferred, from Fig. R1, that the height of position with the greatest potential contribution rate from the traffic source varies depend on the distance between the position and traffic source, as well as the background wind field. The above analysis has been added in Section 3.3.1 and Conclusion in the revised manuscript.**

[Figure]

105 **Figure R1.** The (a) average surface wind and (b) vertical structure of average winds that along the wind direction around the receptor site S2 (line AB) in January 2019.

*3. Another major finding is in l. 329. The authors compare the contribution ratio during westerly wind and easterly wind. The higher ratio during westerly wind is explained by the building distribution. Why does the dense building distribution of upwind of the receptor site lead to a higher contribution ratio? If the authors can explain it, this result should be added in Conclusion and Abstract.*

**Response: Thanks for pointing out it. We did find an error in the manuscript. The potential contribution ratios of traffic source were calculated to be 2.45% and 1.98% for the east and west wind directions, respectively. l. 402 in the revised manuscript should be corrected as: "The lower contribution ratio during westerly winds relatively to that under easterly winds might partially be due to the denser distribution of buildings upwind of the receptor site." We attributed the lower contribution ratio during westerly winds might partially be due to the denser distribution of buildings upwind of the receptor site and added a more detailed explanation in Section 3.3.2 of the manuscript. "Complex building layouts changed the structure of the wind field and thus had an impact on the transport of pollutants. The slow air circulation in dense building areas made it unfavorable for pollutants to be transported. In the windward side of the dense building area, the wind was blocked and diverted to both sides of the building. Pollutants were difficult to transport to the leeward side of the building, where the receptor site was located." The results have been added in the Conclusion and Abstract in the revised manuscript.**

*4. In l. 21, "which is more obvious for high buildings and influencing air pollution transport at the street-level." is not discussed in the text. Is it related to any results of this study?*

**Response: The phenomenon of divergence and convergence of high buildings was not explicitly discussed in the text, and we have added it in Section 3.2.2 of the manuscript: "High-rise buildings have a greater impact on the wind field and cause a strong degree of convergence and divergence. It can be seen that the degree of divergence or convergence around the high-rise building is more significant than those around low buildings in the area (Figs. 6b-3, 6c-3, and 6d-3)."**

*5. One of important issues on the atmospheric modeling using CFD model is the computation time. How have you conducted a simulation on such a large domain in Figure 3 for about one month? Could you briefly explain the technical part in Appendix?*

**Response: It is true that using a CFD model for the atmospheric numerical simulation has the problem of high computational cost. In this study, the RANS is chosen as the CFD pre-processing model, which requires relatively small amount of computational resources. The time step of STAR-CCM+ is set to 60 s, with a maximum of 20 internal iterations in each time step and a parallel computing with 32 CPUs is done on a supercomputer. The simulation error increases with the simulation time. In order to ensure the efficiency and accuracy of the simulation, the month was divided into four time periods to simulate, as shown in Table R1. We have added additional explanation into the Appendix of the manuscript.**

Table R1. The division of each simulation time period and the physical time spent on the simulation

| Simulation start time | Simulation end time | Length of simulation time | Physical time spent |
|---|---|---|---|
| 2018/12/31 00:00:00 | 2019/1/ 09 04:00:00 | 220h | 126.45h |
| 2019/1/08 00:00:00 | 2019/1/ 17 04:00:00 | 220h | 128.33h |
| 2019/1/ 16 00:00:00 | 2019/1/25 04:00:00 | 220h | 128.53h |
| 2019/1/24 00:00:00 | 2019/2/01 08:00:00 | 200h | 117.10h |

Minor comments

*1. l.12, l.86, l.93, and many other lines: the CFD model name should be given.*

**Response: They have been changed to STAR-CCM+ in the manuscript.**

*2. l.26-27: please rewrite the sentence.*

**Response: It has been rewritten in the revised manuscript.**

*3. l.37: why is the term "diffusion" used? Many times, "transport and diffusion" appear in the text (l.203, l.224). "diffusion" is one phenomena of transport.*

**Response: Thanks for pointing it out. We agree with that the diffusion is only one of transport processes. The term "transport and diffusion" has been changed to "transport" in the revised manuscript.**

*4. l.69: remove "In 2006"*

**Response: It has been deleted.**

*5. l.69-70: the sentence is not clear.*

**Response: It has been changed to "Fast and Easter (2006) developed a FLEXPART version that used the WRF model output and was optimized with technical level and output results." in the manuscript.**

*6. l.74-75: the sentence is not clear. What do you mean with "the spatial resolution is affected by the numerical dispersion in the Eulerian model?"*

**Response: Sorry for that unclear description. We mean that, in Eulerian model, the spatial and temporal dispersion should meet the Courant–Friedrichs–Lewy (CFL) condition, otherwise instability will lead to numerical simulations that "blow-up" (Stam, 1999). In solving the Navier-Stokes equations, the biggest difference between Eulerian and Lagrangian method is the solution of the advection term, which stands for transfer of fluid particles between different grids. The Eulerian method is concerned with the fixed points in space, where the physical field values at the fixed points in space vary with time. In solving the advection process, it is limited by the CFL condition, and thus for small spatial resolution, very small time steps have to be taken. The Lagrangian model is not subject to the CFL condition. It has been revised in the manuscript: "Different from Eulerian model, the Lagrangian model is not restricted by the Courant–Friedrichs–Lewy (CFL) condition (Stam, 1999) and thus, the integration process in the Lagrangian model can be maintained with high spatial resolution with acceptable computation efficiency."**

**Reference:**

Stam, J.: Stable Fluids, ACM Trans. Graph., 1999, https://doi.org/10.1145/311535.311548, 1999.

*7. l.96-98: as Major comment 1, explain which data of WRF are used to fill the gap.*

**Response: It has been explained in the reply to Question 1 and Table 1 was added in the revised manuscript to show which elements are filled by WRF results (as shown in Table R2).**

Table R2. The list of variables required to run FLEXPART and the sources of variables.

| variable | Description | Source |
|---|---|---|
| PB | base value of pressure | WRF |
| P | perturbation of pressure | WRF |
| PHB | base value of geopotential | WRF |
| PH | perturbation of geopotential | WRF |
| T | temperature | WRF |
| QVAPOR | specific humidity | WRF |
| MAPFAC_M | map factor | WRF |
| PSFC | surface pressure | STAR-CCM+ |
| U10 | 10 m wind along $x$ axis | STAR-CCM+ |
| V10 | 10 m wind along $y$ axis | STAR-CCM+ |
| T2 | 2 m temperature | WRF |
| Q2 | 2 m dew point | WRF |
| SWDOWN | surface solar radiation (optional) | WRF |
| RAINNC | large scale precipitation (optional) | WRF |
| RAINC | convective precipitation (optional) | WRF |
| HFX | surface sensible heat flux (optional) | STAR-CCM+ |
| U | wind along $x$ axis | STAR-CCM+ |
| V | wind along $y$ axis | STAR-CCM+ |
| W | Cartesian vertical velocity | STAR-CCM+ |

*8. l.116: remove "powerful". Please add the references that the STAR-CCM+ has been used for street-level simulation to show its modeling performance.*

**Response: The term "powerful" has been deleted. Previous studies had found an excellent correlation between STAR-CCM+ simulated and measured values in simulating environmental and meteorological problems at street-level. References show modeling performance of STAR-CCM+ for street-level**

simulation (Santiago et al., 2017; Borge et al., 2018; Jls et al., 2020) have been added to the manuscript.

**Reference:**

215 Borge, R., Jose Luis Santiago, David de la Paz, Fernando Martín, Jessica Domingo, Cristina Valdés, Beatriz Sánchez, Esther Rivas, Ma Teresa Rozas, Sonia Olaechea Lázaro, Pérez, J., and Fernández., a. Á. L.-P.: Application of a short term air quality action plan in Madrid (Spain) under a high-pollution episode - Part II: Assessment from multi-scale modelling, Sci. Total Environ., 635, 1574-1584, https://doi.org/10.1016/j.scitotenv.2018.04.323, 2018.

220 Jls, A., Bse, A., Cq, B., Ddlp, B., Am, A., Fm, A., Rb, B., Er, A., Gm, A., and Ed, A.: Performance evaluation of a multiscale modelling system applied to particulate matter dispersion in a real traffic hot spot in Madrid (Spain) - ScienceDirect, Atmospheric Pollut, 11, 141-155, https://doi.org/10.1016/j.apr.2019.10.001, 2020.

Santiago, J. L., Rafael Borge, Fernando Martín, David de la Paz, Alberto Martilli, Lumbreras, J., and
225 Sánchez, a. B.: Evaluation of a CFD-based approach to estimate pollutant distribution within a real urban canopy by means of passive samplers., Sci. Total Environ., 576, 46-58, https://doi.org/10.1016/j.scitotenv.2016.09.234, 2017.

*9. l.122-123: correct the section title to "3D street-level grid generation". What do you*
230 *mean "geometric model"?*

**Response: It has been corrected. The "3D geometric model" is used to represent the geometry shape of underlying buildings, and is constructed from the real building's shape. It is used as the base data to drive the STAR-CCM+ simulation. It has been explained in the revised manuscript.**

235

*10. l.135 and l.138: correct "grids" to "grid cells". Please don't mix "grid" and "grid cells"*

**Response: It has been corrected.**

240 *11. l.139: I think the equations are not useful to explain the STAR-CCM+ model. Please rewrite this section focusing on the coupling of WRF model to STAR-CCM+.*

**Response: We have removed the equations and added the coupling of WRF model to STAR-CCM+ in Section 2.3.2 of the manuscript.**

245 *12. l.168: why are the turbulence options needed in FLEXPART? Presenting the main idea of the FLEXPART may be helpful. "CBL" can be removed.*

**Response: The turbulence options are needed in FLEXPART to resolve the sub-grid turbulence, since the grid resolution of its host model (such as WRF)**

**is not fine enough. As the STAR-CCM+ could resolve most of the turbulence at very fine gird resolution, the turbulence option of FLEXPART model can be turned off in this research. We have added main idea of the FLEXPART in Section 2.4 of the manuscript and deleted the "CBL".**

*13. l.179: add Wind direction. It is presented in Figure 4.*

**Response: It has been added. The wind direction has been added to Figure 4 in the revised manuscript (as shown in Fig. R2). The corresponding evaluation of wind direction has been also added in the manuscript.**

[Figure]

**Figure R2:** Evaluation of the wind simulation results at the BSC monitoring site (see in Fig. 3a): the simulated, by WRF (blue line) and STAR-CCM+ (red line) model, respectively, and the observed (grey line) hourly near-surface wind speeds (a) and wind directions (b).

15  *14. l.181: correct mean deviation to mean bias.*

**Response: It has been corrected.**

*15. l.189-190: T and RH are underestimated. P and WS (correct W to WS) are overestimated.*

20  **Response: They have been corrected.**

*16. l.192: explain what is the significance test at least in Appendix.*

**Response: Significance test is used to determine the significance of the results in relation to the null hypothesis, with a p-value, or probability value describing how likely the data would have occurred by random chance (i.e. that the null hypothesis is true). A p-value less than 0.05 (typically $\leqslant$ 0.05) is statistically significant. It indicates strong evidence against the null hypothesis, as there is less than a 5% probability the null is correct. Following your suggestion, we have detailed it in Appendix in the revised manuscript.**

*17. l.197-199: This sentence does not have proof. Please remove it.*

**Response: It has been deleted.**

*18. l.206: the averaged wind speed of 0.92 m/s is very low speed. Models cannot easily reproduce the low speed. Why is the observed wind speed very low? Is it a typical value on the simulation domain in winter?*

**Response: We have compared the observed data for January 2019 at various meteorological stations in Zhengzhou City. Table R3 shows the location of each meteorological station and the average wind speed. The overall average wind speed in January 2019 in Zhengzhou City was 1.06 m s⁻¹, and the average wind speed at the Bank School City monitoring site was 0.92 m s⁻¹, which is a typical wind speed value for the simulated domain in winter.**

Table R3. The location of each meteorological station and the average wind speed.

| number | Latitude and longitude coordinates | Average wind speed |
|---|---|---|
| 1 | (34.7274 N, 113.7493 E) | 0.92 m s⁻¹ |
| 2 | (34.73506 N, 113.6457 E) | 0.92 m s⁻¹ |
| 3 | (34.7466 N, 113.6876 E) | 1.32 m s⁻¹ |
| 4 | (34.76117 N, 113.6883 E) | 0.61 m s⁻¹ |
| 5 | (34.78245 N, 113.6567 E) | 1.51 m s⁻¹ |
| 6 | (34.81151 N, 113.6948 E) | 1.48 m s⁻¹ |
| 7 | (34.83267 N,113.5453 E) | 0.72 m s⁻¹ |

*19. l.218: This sentence does not have proof. Please remove it.*

**Response: It has been deleted.**

*20. l.235: How is the influence of buildings on the wind field estimated? How do you know it is diminished?*

**Response: On near-surface level (at 5m height), the buildings cluster is dense. The airflow is affected by the layout of the building, rendering the wind directions differed notably inside the block (Fig. 5a). As the density of buildings gradually decreases with the increases of height, the phenomenon diminish, reflecting by the relatively more consistent wind fields at 40 m (Fig. 5b). Thanks for your suggestion, we have detailed it in the manuscript.**

*21. l.299-l.302: the authors may remind the results in Section 3.2 to link the differences in the wind field to potential contribution ratio.*

**Response: Thanks for pointing this out. A note in Section 3.3.1 of the manuscript has been added: "From the average wind field in January 2019 (Fig.R3), it can be seen that the winds are influenced by high-rise buildings around the S1, resulting in a change in transport path of pollutants and thus, making pollutants difficult to reach the S1 site. However, for S2 site, the winds were less influenced by the buildings and pollutants were more easily transported there. "**

[Figure]

**Figure R3.** The average wind field in January 2019.

*22. Figure 4: Why have you not compared WRF wind direction?*

**Response: We have added the WRF wind direction to Figure 4 of the revised manuscript and evaluated the simulated wind direction by hit rates (HR) (Schlünzen and Sokhi, 2008). With desired accuracy between ±45◦, the HR were calculated at 63 and 51 % for STAR-CCM+ and WRF, respectively, indicating that variations in wind direction were basically captured a better performance for STAR-CCM+ simulations.**

**Reference:**

Schlünzen, K. H. and Sokhi, R. S.: Overview of Tools and Methods for Meteorological and Air Pollution Mesoscale Model Evaluation and User Training, Joint report by WMO and COST 728, WMO/TD-No. 1457, Geneva, Switzerland, 2008, 2008.

*23. Figure 5: Please make figures bigger and explain what is the divergence.*

**Response: The figures have been enlarged and the divergence has been explained in Appendix. The divergence is a quantity that describes the degree to which air converges from its surroundings to a point or flows away from a point. It is used to describe the intensity of divergence and convergence at locations in space. The formula is as follows.**

$$div \ \mathrm{v} = \nabla \bullet \mathrm{v} = \frac{\partial u_i}{\partial x_i} = \frac{\partial u}{\partial x} + \frac{\partial v}{\partial y} + \frac{\partial w}{\partial z},$$

**where *u*, *v* and *w* are the components of the wind in the *x*, *y* and *z* direction, respectively. When the $div \ \mathrm{v} < 0$, the location is convergence; when the $div \ \mathrm{v} > 0$, the location is divergence.**

*24. Figure 7: Explain what is the density distribution in the text.*

**Response: In this study, the density distribution is the number of particles that have stayed in the space of 10 m × 10 m in horizontal direction and 5 m from the road surface to above in vertical direction during January 2019. We have explained it in Figure 7 of the manuscript.**

*25. Table 3: Are the potential contribution ratio the averaged value of all wind directions?*

**Response: In Table 4 of the revised manuscript, the listed potential contribution is the overall simulation result of all wind directions (without dividing the wind directions), not the averaged value. It has been stated in the revised manuscript. In addition, we calculated the potential contribution of traffic sources separately under different wind directions in section 3.3.2 of the manuscript.**

**Referee #2**

*The authors presented an interesting work that integrates the Weather Forecasting Model (WRF), Computational Fluid Dynamics (CFD) and Flexible Particle (FLEXPART) to study local circulations at a neighbourhood scale and the potential contribution of one street as a source of air pollution. The authors analyzed local wind circulation under different regional wind regimes. The methods used by the authors are robust and the meteorology simulated shows good agreement as shown in figure 4. However, conclusions, as stated in lines 338-339, authors say: "In general, the S-TRACK system is effective in simulating meteorological and air pollution problems" has no strong basis. This is because the is no real emissions and only one street was evaluated. Specifically:*

Specific comments

*1) in the abstract, the authors state: "The results of the study are helpful to understand the characteristics of wind environment and effect of traffic emissions in the area...", however, there are no traffic emissions in the study.*

> **Response: Thank you sincerely for pointing it out. In the current study, we did not have the detailed information about the traffic emissions from the road, so we assumed that the road traffic emissions are uniformly distributed and with identical intensity. During the backward trajectory simulation, the particles as long as passing within 5 m height above the road is considered to be a potential contribution from the road emissions to the receptor site. Using the methods of residence time analysis (RTA) to determine the magnitude of the potential contribution, the RTA is expressed as follows.**

$$R_{i,j} = \frac{\tau_{i,j}}{t},$$

> **where $R_{i,j}$ indicates the contribution ratio of the grid $(i, j)$ to receptor; $\tau_{i,j}$ means the residence time in the grid $(i, j)$ and $t$ means the total residence time in all grids. There is some uncertainty in this approach. Your suggestion is much appreciated, and we have modified the section as: "The established system and the results can be used to understand the characteristics of urban wind environment and to help the air pollution control planning in urban areas." In addition, we also provide additional clarification for this situation**

in Section 3.3.

*2) The authors presented the potential impacts of only one street. however, we do not know anything about that street, type of traffic, the diurnal cycle of traffic by type of vehicle and as shown in Figures 3a, there are many streets surrounding the area. For instance, Figure 7b shows that the street might be important for site S2, but this site is closer to a perpendicular street (east), as shown in figure 3a. In a summary, despite that the results are interesting, more work is needed to provide the basis for the conclusions.*

**Response: Indeed, there are many roads around the area that may have impacts on the receptor sites. According to your suggestions, we have selected four main roads in the region as shown in revised Figure R4 to analyze the potential impact characteristics of traffic sources on receptor sites. The widths of roads (including greenery and open space) R1-R4 are about 45, 33, 20 and 18 meters, respectively. We have done our best to collect the information of the roads. During January 2019, the average hourly traffic volumes are about 2300 (R1), 490 (R2), 400 (R3) and 90 (R4) cars, respectively. We assume that the road emissions are uniformly emitted and with identical intensity. The magnitude of the potential contribution of the four main roads of the sub-domain B to each receptor site is shown in the Table R4. Following your suggestions, we have analyzed the potential contribution of the four roads to the receptor sites (S2-S8) at different heights separately in Section 3.3.1 in the revised manuscript, and found that road R1 had the largest potential contribution to the receptor sites. Accordingly, in the subsequent analysis in the manuscript, we only considered the contribution of R1 to different positions (i.e., S1 and S2 sites).**

[Figure]

**Figure R4:** The computational domain of the CFD model, with magenta dots (S1 and S2) indicating the receptor sites and Orange line indicating the main roads in Sub-domain B.

Table R4. Locations of receptor sites and the corresponding potential contribution ratios.

| Receptor site | Location (x, y, z) | potential contribution ratio | | | | |
|---|---|---|---|---|---|---|
| | | R1 | R2 | R3 | R4 | All |
| S1 | (-3200 m, -1420 m, 2 m) | 1.81% | - | - | - | - |
| S2 | (-2500 m, -1300 m, 2 m) | 2.38% | 0.18% | **1.32%** | 0.16% | 4.05% |
| S3 | (-2500 m, -1300 m, 5 m) | 2.57% | 0.29% | 1.28% | 0.10% | 4.25% |
| S4 | (-2500 m, -1300 m, 10 m) | 2.71% | 0.32% | 1.18% | 0.12% | 4.33% |
| S5 | (-2500 m, -1300 m, 15 m) | **2.98%** | 0.27% | 1.22% | 0.20% | **4.67%** |
| S6 | (-2500 m, -1300 m, 20 m) | 2.75% | 0.37% | 1.09% | 0.17% | 4.38% |
| S7 | (-2500 m, -1300 m, 40 m) | 2.30% | 0.39% | 0.70% | 0.25% | 3.64% |
| S8 | (-2500 m, -1300 m, 50 m) | 1.94% | 0.57% | 0.68% | 0.36% | 3.55% |

*3) Figure 1 shows that the results are interpolated but the authors do not show the methods and reasons for this interpolation.*

**Response: In this study, the method, used to interpolate the simulation results of WRF and STAR-CCM+ into a 10m×10m regular grid, is the nearest neighbor interpolation method. The nearest neighbor method is based on a comparison of the distribution of distances between a point and the nearest neighboring points of a set of randomly distributed data. The principle of the nearest neighbor interpolation is to select the value at the nearest location to**

the inserted grid as the value of the grid point. It is effective for interpolating data with uniformly spaced distribution (e.g., WRF simulation results) as well as tightly integrated data (e.g., STAR-CCM+ simulation results). Compared with other interpolation methods, the method also has the advantages of small computational effort, simple algorithm, fast operation speed, and easy implementation(Yang et al., 2004). We have added the description in Section 2.1 in the revised manuscript.

**Reference:**

Yang, C. S., Kao, S. P., Lee, F. B., and Hung, P. S.: Twelve different interpolation methods: A case study of Surfer 8.0, In: Proceedings of ISPRS Congress, 20, 778-785, 2004.

Minor issues:

*1. The authors simulated between December 30 2018 and January 31, 2019. Why this period?*

**Response: An El Niño event occurred in January 2019 according to the Oceanic Niño Index (ONI). The occurrence of El Niño generally favors a warm winter and weak winter winds. Winds are one of the most important meteorological factors affecting the transport of pollutants. Therefore, it is more meaningful to study the wind environment and pollutant transport during this time period. We have explained the reason for choosing January 2019 as the simulation period in the manuscript.**

*2. Line 69: add space before Fast and remove space in "( 2006"*

**Response: It has been corrected.**

*3. Lines 84-88. Only one phrase for a 5 lines paragraph. Each paragraph must be formed by at least three phrases, introduction, body and conclusion.*

**Response: We have rewritten the paragraph in the revised manuscript.**

*4. Line 124: Why do you need drone aerial photography? I think it is very good, but I'm surprised that you are not trying other sources.*

**Response: Drone aerial photography technology can quickly obtain all kinds of basic data such as the geometric shape of urban buildings, roof height and**

vector data of the top of buildings with high resolution, high timeliness and accuracy. The information can be used to build a more realistic 3D model. Considering other methods, such as processing various scale line planning topographic maps or satellite remote sensing maps to collect urban building data, there are shortcomings such as poor visibility or high cost. Therefore, the drone aerial photography technique is chosen to obtain the city 3D modeling. We have added the description in the revised manuscript.

*5. Lines 162-164: "This... FLEXPART", this phrase is redundant.*

**Response: It has been deleted.**

*6. Lines 172: What was the number of particles released?*

**Response: In the course of simulation, 5 tracer particles are released per hour, and the total number of particles released was 3590 tracer particles. We have detailed it in the manuscript.**

7. Lines 197-198: Simulating well the PBL is important to the dispersion of pollutants. How are the results? The authors can include more results in the supplementary material.

**Response: Thanks for your suggestion. The bulk Richardson number (Ri) method was taken to estimate the BLH base on the sounding data of Zhengzhou. Ri is expressed as:**

$$R_i(z) = \frac{(g/\theta_{vs})(\theta_{vz} - \theta_{vs})(z - z_s)}{(u_z - u_s)^2 + (v_z - v_s)^2 + (bu_*^2)},$$

**where $z$ means height above ground, $s$ means the surface, $g$ means the acceleration of gravity, $\theta_v$ means the virtual potential temperature, $u$ and $v$ mean the component of wind speed, and $u_*$ means the surface friction velocity. $u_*$ can be ignored here due to it is small relative to the wind shear (Vogelezang and Holtslag, 1996). Previous theoretical and laboratory studies suggested that when Ri is smaller than a critical value (~0.25), the laminar flow becomes unstable (Stull, 1988). Therefore, the lowest level z at which the**

**interpolated Ri exceeds the critical value of 0.25 is referred to as PBLH in this study, which is referred to the criterion used by Seidel et al. (2012). The R value is 0.57, passed the 99% significance test. It can be seen from figure R5 that the variation of boundary layer height is generally captured.**

[Figure]

**Figure R5.** Time series of the observed (black) and simulated (red) PBLH at 8:00 and 20:00 Beijing time (BJT) in Zhengzhou sounding site.

**Reference:**

Seidel, D. J., Zhang, Y., Beljaars, A., Golaz, J. C., Jacobson, A. R., and Medeiros, B.: Climatology of the planetary boundary layer over the continental United States and Europe, Journal of Geophysical Research Atmospheres, 117, https://doi.org/10.1029/2012JD018143, 2012.

Stull, R. B.: An Introduction to Boundary Layer Meteorology, Springer Netherlands, Dordrecht,1988.

Vogelezang, D. and Holtslag, A.: Evaluation and model impacts of alternative boundary-layer height formulations, Bound.-Lay. Meteorol., 81, 245-269, https://doi.org/10.1007/BF02430331, 1996.

*8. Lines 219: This line seems incomplete.*

**Response: Another referee thought this sentence does not have proof and should be deleted. Therefore, we have removed it from the manuscript.**

*9. Section 3.2.1. There are many parts where English needs to be improved with a more technical language. For instance line 232 authors say: "The wind is larger", maybe more intense is more appropriate. Lines 234-237 needs edit/improvement.*

**Response: Thank you very much for pointing this out. We have improved the English of the whole manuscript, including the Section 3.2.1. Please see in the marked-up manuscript.**

*10. Line 240: change climb or fall subsidence or other.*

**Response: "climb or fall" has been changed to "rise or subsidence".**

*11. Line 244: Change pile up for other.*

**Response: "pile up" has been changed to "accumulate".**

5 *12. Line 261: Line seems repeated.*

**Response: It has been deleted.**

*13. Figure 5 showing streamlines and divergence. Consider merging streamlines with shaded values for divergence. In that way, it would be easier to see diffluence.*

10 **Response: Thank you for the suggestion. The streamlines and divergence has been merged in Figure 5 in the revised manuscript.**

[Figure]

**Figure 5.** The simulated winds and divergence at the near-surface averaged in January 2019 in the sub-domain A.

---

## Referee Report (RR1)

Referee's comments on Development and application of a street-level meteorology and pollutant tracking system (S-TRACK)

1. L.28: Add the result for the conclusion for Section 3.3 after line 28-29
2. L.38: Check the order of in-text citations. Here is the guideline of ACP

In terms of in-text citations, the order can be based on relevance, as well as chronological or alphabetical listing, depending on the author's preference.

3. L.51 : pre-processing can be removed.
4. L.55 : CFD technology, CFD approach, CFD simulation
5. L.65: give the full name of NO2 and O3 because they appear in the text for the first time.
6. L.75: give the full name of ECMWF and NCEP.
7. L.81: give the full name of NOx.
8. L.81: correct WRF-LES to WRF
9. L.125: correct base -> based
10. L.126: correct refine -> refined
11. L.135: correct grids -> grid cells. There are other places to check.
12. L.149: correct the citation and also in the reference list: Jls et al. to Santiago et al.
13. L.167: correct Figs -> Fig
14. L.192: If you do not use Turbulence option in FLEXPART, I think you don't have to explain the option. You may remove the paragraph from "plus random" to line 198.
15. L.203: remove "tracer particles in the course of simulation".
16. L.220: add the minus sign, 1.86K to -1.86K and 5.95% to -5.95%
17. L.222: correct as "significance test (see Appendix 2)"
18. L.234: rewrite the sentence.
19. L.244: define the hit rates (HR)
20. L.307-L.311: In abstract, you explain that an identical traffic volume is used. Is it true? You mention the average traffic volumes for R1 to R4 in L.308. "identical intensity" is same to identical traffic volume?
21. L.312-322: I understand that RTA is the potential contribution ratio. RTA is computed using the residence time in the grid cell (i, j) of the particles passing within 5 m height above the road. Is it right? Using the acronym PCR for potential contribution ratio may be helpful.
22. L.324: "potential impact" and "potential contribution" are confusing. You may improve the manuscript using one of them.
23. L.337: explain what is the density distribution in the text instead of Figure 7.
24. L.353: I think higher potential contribution ratio from R1 than R3 may be due to higher traffic volumes (see L.308) at R1 if you use different traffic volumes (see my question 20).
25. L.409: Please write the conclusion from Section 3.3 (L.415-L.421) and then that from Section 3.2 according to the order in the text.
26. L. 702: Correct Table R4 to Table 4

---

## Author Response (AR2)

Dear Editor and referees,

We are very grateful for your time and attentions on this manuscript. Please find below our itemized responses to the referees' comments. We have addressed all the comments raised by the referee and incorporated them in the revised manuscript.

Thank you for your consideration.

Sincerely,

Sunling Gong
* * *
*Referee's comments on Development and application of a street-level meteorology and pollutant tracking system (S-TRACK)*

*1. L.28: Add the result for the conclusion for Section 3.3 after line 28-29*

**Response: It has been added.**

*2. L.38: Check the order of in-text citations. Here is the guideline of ACP*

*In terms of in-text citations, the order can be based on relevance, as well as chronological or alphabetical listing, depending on the author's preference.*

**Response: The citations in-text have been arranged in alphabetical listing.**

*3. L.51: pre-processing can be removed.*

**Response: It has been removed.**

*4. L.55: CFD technology, CFD approach, CFD simulation*

**Response: "CFD technology" has been changed to "CFD simulation".**

*5. L.65: give the full name of NO2 and O3 because they appear in the text for the first time.*

**Response: They have been given the full names.**

*6. L.75: give the full name of ECMWF and NCEP.*

**Response: They have been given the full names.**

*7. L.81: give the full name of NOx.*

**Response: It has been given the full names.**

*8. L.81: correct WRF-LES to WRF*

**Response: It has been corrected.**

*9. L.125: correct base -> based*

**Response: It has been corrected.**

*10. L.126: correct refine -> refined*

**Response: It has been corrected.**

*11. L.135: correct grids -> grid cells. There are other places to check.*

**Response: They have been corrected.**

*12. L.149: correct the citation and also in the reference list: Jls et al. to Santiago et al.*

**Response: It has been corrected.**

*13. L.167: correct Figs -> Fig*

**Response: It has been corrected.**

*14. L.192: If you do not use Turbulence option in FLEXPART, I think you don't have to explain the option. You may remove the paragraph from "plus random" to line 198.*

**Response: Thanks for pointing this out. It has been removed.**

*15. L.203: remove "tracer particles in the course of simulation".*

**Response: It has been removed.**

*16. L.220: add the minus sign, 1.86K to -1.86K and 5.95% to -5.95%*

**Response: Thanks for pointing this out. they have been added.**

*17. L.222: correct as "significance test (see Appendix 2)"*

**Response: It has been corrected.**

*18. L.234: rewrite the sentence.*

**Response: It has been rewritten.**

*19. L.244: define the hit rates (HR)*

**Response: Its definition has been explained in Appendix in the revised manuscript.**

*20. L.307-L.311: In abstract, you explain that an identical traffic volume is used. Is it true? You mention the average traffic volumes for R1 to R4 in L.308. "identical intensity" is same to identical traffic volume?*

**Response: In this study, we used the same traffic density instead of traffic volume. We have corrected in abstract of the revised manuscript.**

*21. L.312-322: I understand that RTA is the potential contribution ratio. RTA is computed using the residence time in the grid cell (i, j) of the particles passing within 5 m height above the road. Is it right? Using the acronym PCR for potential contribution ratio may be helpful.*

**Response: RTA is computed using the residence time in the grid cell (i, j) of the particles passing within 5 m height above the road. However, RTA is not the potential contribution ratio, is a method to calculate the potential contribution**

ratio. We have used PCR as the acronym for potential contribution ratio in the revised manuscript.

*22. L.324: "potential impact" and "potential contribution" are confusing. You may improve the manuscript using one of them.*

**Response: The term "potential contribution" has been unified using in the revised manuscript.**

*23. L.337: explain what is the density distribution in the text instead of Figure 7.*

**Response: We have added the explanation of the density distribution to the text and removed the explanation in Figure 7.**

*24. L.353: I think higher potential contribution ratio from R1 than R3 may be due to higher traffic volumes (see L.308) at R1 if you use different traffic volumes (see my question 20).*

**Response: In this study, we assumed that the traffic intensity of the roads is consistent. Therefore, the potential contribution ratio of R1 is higher than that of R3, due to higher the road width of R1. Also, to avoid misleading the readers' judgment, we have removed the information on the average road traffic volume in the revised manuscript.**

*25. L.409: Please write the conclusion from Section 3.3.1 (L.415-L.421) and then that from Section 3.3.2 according to the order in the text.*

**Response: It has been changed. The conclusion of section 3.3.1 was written before the conclusion of 3.3.2 in the revised manuscript.**

*26. L. 702: Correct Table R4 to Table 4*

**Response: It has been corrected.**